 

# Developmental evolution of the forebrain in cavefish, from natural variations in neuropeptides to behavior

Alexandre Alié[†], Lucie Devos[†], Jorge Torres-Paz[†], Lise Prunier, Fanny Boulet, Maryline Blin, Yannick Elipot, Sylvie Retaux*

Paris-Saclay Institute of Neuroscience, Université Paris Sud, CNRS UMR9197, Université Paris-Saclay, Avenue de la terrasse, Gif-sur-Yvette, France

**Abstract** The fish *Astyanax mexicanus* comes in two forms: the normal surface-dwelling and the blind depigmented cave-adapted morphs. Comparing the development of their basal forebrain, we found quantitative differences in numbers of cells in specific clusters for six out of nine studied neuropeptidergic cell types. Investigating the origins of these differences, we showed that early Shh and Fgf signaling impact on the development of NPY and Hypocretin clusters, via effect on Lhx7 and Lhx9 transcription factors, respectively. Finally, we demonstrated that such neurodevelopmental evolution underlies behavioral evolution, linking a higher number of Hypocretin cells with hyperactivity in cavefish. Early embryonic modifications in signaling/patterning at neural plate stage therefore impact neuronal development and later larval behavior, bridging developmental evolution of a neuronal system and the adaptive behavior it governs. This work uncovers novel variations underlying the evolution and adaptation of cavefish to their extreme environment.
DOI: https://doi.org/10.7554/eLife.32808.001

*For correspondence:
retaux@inaf.cnrs-gif.fr

[†]These authors contributed equally to this work

**Competing interests:** The authors declare that no competing interests exist.

## Introduction

The secondary prosencephalon of the vertebrate forebrain, comprising the telencephalon, the optic/preoptic region and the hypothalamus, develops from the anterior neural plate. By the end of gastrulation, the neural plate is already patterned and the regional fate of its antero-posterior and medio-lateral domains is specified, as a result of the concerted action of diffusible morphogen molecules that emanate from secondary organizers (reviewed in [*Cavodeassi and Houart, 2012*]). Endo-mesodermal tissues (prechordal plate and notochord) located ventral to the neural plate secrete Nodal/TGFβ and Sonic Hedgehog (Shh) molecules necessary for induction of ventral forebrain and maintenance of hypothalamic fate (*Chiang et al., 1996*; *Kiecker and Niehrs, 2001*; *Mathieu et al., 2002*). The anterior-most neural plate border, named ANB then ANR (anterior neural border and ridge) secrete Wnt inhibitors and Fibroblast Growth Factors that are required for the establishment of telencephalic fate (*Houart et al., 2002*; *Houart et al., 1998*) and its patterning (*Miyake et al., 2005*; *Shanmugalingam et al., 2000*; *Shimogori et al., 2004*; *Walshe and Mason, 2003*), respectively. Thus, the elaboration of the vertebrate forebrain depends on the tight spatial and temporal regulation of relatively few morphogenetic signals. Changes in these early events have the potential to modulate brain organization, notably the relative size of different brain regions (*Hinaux et al., 2016*; *Sylvester et al., 2010*; *Sylvester et al., 2013*) (reviewed in [*Rétaux et al., 2013*]).

The evolution of brain development and its behavioral consequences is a major topic to understand how vertebrates colonize novel environments. *Astyanax mexicanus* is a model of choice to tackle this question (*Jeffery, 2008*; *Jeffery, 2009*; *Rétaux et al., 2016*). This teleost fish exhibits two morphotypes: a surface-dwelling form (thereafter designated as SF) that inhabits South and

Central America rivers, and a cave-dwelling form (CF) that consists in multiple populations living in the total and permanent darkness of Mexican caves (*Mitchell et al., 1977*). These two forms have split from a surface fish-like ancestor less than 30,000 years ago (*Fumey et al., 2017*). During this time, cavefish have evolved regressive traits - the most spectacular being the loss of eyes and pigmentation - but they also evolved several constructive traits such as a larger jaw, more taste buds and neuromasts, or larger olfactory epithelia (reviewed in [*Rétaux et al., 2016*]).

Inside the cavefish brain, patterning modifications have been described as well and are the consequences of subtle modifications in the early expression of morphogens during evolution. Enlarged ventral midline Shh expression during gastrulation and neurulation (*Yamamoto et al., 2004*) results in a larger hypothalamic region, presumably via extension of *Nkx2.1a*, *Lhx6* and *Lhx7* transcription factor expression domains and increased cell proliferation (*Menuet et al., 2007*; *Rétaux et al., 2008*). Early expansion of Shh is also responsible for a heterochrony in the onset of *Fgf8* expression in the ANR (*Pottin et al., 2011*). Fgf8 is turned on at 10hpf in CF ANR, while its expression starts 2 hrs later in SF. In turn, Fgf8 maintains *Shh* expansion in the CF developing forebrain basal plate, with consequences on neural plate patterning at the onset of neurulation: the expression domains of transcription factors (TFs) such as *Lhx2*, *Lhx9*, *Pax6*, or *Zic1* are different in SF and CF at neural plate stage and related differences in fate maps have been reported (*Hinaux et al., 2016*; *Pottin et al., 2011*; *Strickler et al., 2001*). In particular, cell populations that give rise to the ventral quadrant of the retina in SF seem to be allocated to the dorsal retina or to the hypothalamus in CF. Accordingly, early manipulation of Fgf signaling in CF (thus mimicking the SF situation) is able to restore the ventral quadrant of the retina (*Pottin et al., 2011*). Taken together, these studies highlight the morphogenetic consequences of small spatio-temporal changes in early Shh and Fgf signaling in cavefish. Similar variations in early Wnt signaling in distinct ecotypes of cichlid fish result in the development of brains with large or small telencephalon, thalamus and tectum (*Sylvester et al., 2010*; *Sylvester et al., 2013*). Thus, natural variations of forebrain patterning by signaling modulations may be a widespread mechanism for forebrain evolution.

The hypothalamus is a hub that integrates central and peripheral signals and elicits multiple neuroendocrine and homeostatic responses. Hypothalamic circuits control behaviors such as locomotor activity, sleep/wake cycles and food intake, but also responds to metabolic state of the body (level of adiposity and blood glucose) by directly acting on peripheral organs to control energy expenditure ([*Loh et al., 2015*; *Myers and Olson, 2012*; *Tsujino and Sakurai, 2013*] for recent reviews). *Astyanax* cavefish has evolved multiple behaviors of putative adaptive value to live in cave environment where survival challenges reside in finding food and interacting with congeners in permanent darkness. Several of these behaviors are typically controlled by the hypothalamus: CF show enhanced foraging traits and increased locomotor activity (*Elipot et al., 2013*; *Yoshizawa et al., 2015*), sleep loss (*Duboué et al., 2011*) and, for some populations, enhanced food intake and high fat content (*Aspiras et al., 2015*) when compared to their surface conspecifics.

The behaviors described above are under the control of different, but neighboring neuronal populations in the hypothalamus of vertebrates, including fish (*Herget et al., 2014*; *Löhr and Hammerschmidt, 2011*; *Machluf et al., 2011*; *Matsuda et al., 2012a2012*; *Matsuda et al., 2012b*). Here, we have first compared and interpreted in details the developmental neuroanatomy of nine neuropeptidergic cell types in the hypothalamus and preoptic region of SF and CF embryos and larvae, highlighting specific differences in neuron numbers between the two morphs. We then established causal relationships between early signaling modifications, regional patterning and cell specification processes in cavefish. We further showed that Lhx9 and Lhx7 transcription factors serve as 'relays' and are involved in the observed changes in Hypocretin and NPY neurons in cavefish, respectively. Finally we provide evidence that such developmental evolution in hypothalamic neuronal networks affects the behavior of cavefish larvae.

## Results

### Anatomical interpretation

The morphogenetic movements of the early neural plate and tube are complex. Recently several novel interpretations of the developmental neuroanatomy of the vertebrate secondary prosencephalon have been put forward (*Figure 1—figure supplement 1*). In tetrapods, the updated prosomeric

model aims at proposing causal explanations of the hypothalamic and preoptic area structures, by the effects of antero-posterior and dorso-ventral signaling (*Puelles and Rubenstein, 2015*). Two transverse hypothalamo-telencephalic prosomeres, hp1 and hp2, encompassing the four dorso-ventral domains of the neuroepithelium (roof, alar, basal, and floor plates) have been proposed, supported by genoarchitecture and embryological data (*Figure 1—figure supplement 1A*). In zebrafish, a morphogenetic interpretation of the development of the optic/pre-optic region suggests the existence of three morphogenetic units based on centrifugal neurogenesis patterns rather than on gene expression boundaries –the telencephalon, the optic recess region (ORR) and the hypothalamus-, which helps to resolve some inconsistencies between tetrapod and teleost basal forebrain (*Affaticati et al., 2015*)(*Figure 1—figure supplement 1B*). Finally in zebrafish also, a homology relationship was proposed between the teleost 'neurosecretory pre-optic area' or NPO (located in the ORR as defined by Affaticati and colleagues, or in the preoptic region/PO according to the classical nomenclature) and the mammalian paraventricular nucleus (PVN), which belongs to the mammalian alar hypothalamus (*Herget et al., 2014*; *Herget and Ryu, 2015*)(*Figure 1—figure supplement 1C*). Here, our analyses take into account both the crucial interpretation along the neural axis and according to the alar or basal plate nature of the neuroepithelium, and the peculiarities of the teleost forebrain.

With the aim to precisely document cavefish basal forebrain developmental evolution, we have first compared the ontogenesis of neuropeptidergic cell groups, their organization and their size, between SF and CF embryos and larvae originating from the Pachón cave, at three critical stages of forebrain development (see Materials and methods, *Figure 1—figure supplement 3* and *Figure 1—videos 1–4*).

## More NPY, Hypocretin and AgRP neurons in developing cavefish

Among the nine studied neuropeptides, NPY, Hcrt and AgRP are expressed by more neurons in CF than SF in our regions of interest (*Figure 1*). The results are summarized in *Figure 1U*, and the raw cell counts are provided as *Supplementary file 1*.

Neuropeptide Y (NPY) is expressed from 24 hpf in a medial and superficial hypothalamic cell group, which progressively shifts posteriorly (36 hpf) and clusters in the mammillary region of the hypothalamus (ventral along the neural axis) at 84 hpf (*Figure 1A–F' and U*). We interpret this cell group as part of the *saccus vasculosus* (see discussion). It is located in the hypothalamic basal plate and may belong to the newly proposed acroterminal domain of the secondary prosencephalon (*Puelles and Rubenstein, 2015*)(*Figure 1—figure supplement 1A*). This group of NPY neurons is significantly more numerous in CF than in SF at all investigated stages (*Figure 1G and U*). From 36 hpf, then 84 hpf, NPY is expressed in many additional brain regions (*Figure 1C–F'*). This includes two symmetrical and large cell groups located in the hypothalamus (basal plate) extending dorsally to the region around the post-optic commissure (poc) at the border of the ORR (*Figure 1E–F'*). In this region as well, CF possess about twice more NPY neurons than SF (*Figure 1H and U*). In contrast to these basal regions, the NPY population located in the telencephalon (alar plate) from 36 hpf onwards was similar in the two morphs and a cluster located in the medial part of the dorsal ORR, around the anterior commissure, contained more cells in SF (*Figure 1C–F' and U*). The NPY cells in the optic tectum (alar mesencephalon) were more numerous in SF than in CF (*Figure 1E'F' and U*; cell counts were not performed, but the difference is obvious).

Hypocretin is expressed as two symmetrical groups in the hypothalamus basal plate (Shh-positive, see below and Figure 5H), in a posterior and dorsal position along the brain axis that may topographically correspond to the peduncular (posterior) hypothalamus (*Puelles and Rubenstein, 2015*). Hcrt expression is turned on at 18hpf (*Figure 1—figure supplement 2*) and remains similar in pattern at later stages (*Figure 1I–N'*). Surprisingly, the number of Hcrt neurons raises and is maximal at 24 hpf (average 27 neurons in CF, 17 neurons in SF) and slightly decreases thereafter. At all stages between 18 hpf and 84 hpf, more Hcrt neurons were found in CF than in SF (*Figure 1O and U*, *Figure 1—figure supplement 3* and *Figure 1—videos 1–4*.

AgRP onset of expression is later than NPY and Hcrt: it is first expressed at 36 hpf in cell clusters located in the tuberal hypothalamus, and then becomes distributed in two lateral elongated patches (*Figure 1P–S'*). No differences are detected regarding AgRP neuron abundance between the two *Astyanax* morphs at 24 hpf and 36 hpf, but at 84 hpf CF possess slightly but significantly more AgRP neurons than SF (*Figure 1T and U*).



**Figure 1.** Comparative development of NPY, Hypocretin and AgRP neurons in SF and CF. (A–S') Photographs of embryonic brains after in situ hybridization for NPY, Hypocretin and AgRP at 24, 36, and 84 hpf. The stages, the lateral or ventral orientations, and the probes are indicated. Red squares and blue squares indicate peptidergic clusters with higher numbers of neurons in CF or in SF, respectively. For this and the following figures, raw data including the number of embryos examined are given in *Supplementary file 1*. (G, H, O, T) Quantification and time-course of cell numbers in

*Figure 1 continued on next page*

*Figure 1 continued*

specific clusters. Mann-Whitney tests. (**U**) Anatomical interpretation of peptidergic patterns and time-course of appearance. Hcrt (stars), NPY (circles) and AgRP (triangles) neurons are reported on schematic embryonic brains, in ventral or lateral views. A color code indicates higher numbers of neurons in CF (red) or in SF (blue), or equivalent numbers (grey). Black clusters were not counted. ac, anterior commissure; hyp, hypothalamus; mam; mamilary hypothalamus; or, optic recess; orr, optic recess region; ot, optic tectum; pit, pituitary; poc, post-optic commissure; tel, telencephalon; tub, tuberal hypothalamus; sv, *saccus vasculosus*.

DOI: https://doi.org/10.7554/eLife.32808.002

The following video and figure supplements are available for figure 1:

**Figure supplement 1.** Models and interpretations of prosencephalic development.
DOI: https://doi.org/10.7554/eLife.32808.003
**Figure supplement 2.** Onset of expression of Hcrt and POMCb around 18hpf.
DOI: https://doi.org/10.7554/eLife.32808.004
**Figure supplement 3.** Comparison of colorimetric versus fluorescent in situ hybridization results.
DOI: https://doi.org/10.7554/eLife.32808.005
**Figure 1—video 1.** Comparison of colorimetric *versus* fluorescent in situ hybridization results.
DOI: https://doi.org/10.7554/eLife.32808.006
**Figure 1—video 2.** Comparison of colorimetric *versus* fluorescent in situ hybridization results.
DOI: https://doi.org/10.7554/eLife.32808.007
**Figure 1—video 3.** Comparison of colorimetric *versus* fluorescent in situ hybridization results.
DOI: https://doi.org/10.7554/eLife.32808.008
**Figure 1—video 4.** Comparison of colorimetric *versus* fluorescent in situ hybridization results.
DOI: https://doi.org/10.7554/eLife.32808.009

## Less POMCa, POMCb and AVT neurons in developing cavefish

Three of the studied neuropeptides are expressed by less neuron in CF than SF: POMCa, POMCb, and AVT (*Figure 2*). The results are summarized in *Figure 2V* and the raw cell counts are provided as *Supplementary file 1*.

POMCa is first expressed at 36 hpf in a few cells at the dorsal border of the hypothalamus basal plate. The POMCa pattern then takes the shape of two elongated stripes in the tuberal part of the terminal (or anterior) hypothalamus at 84 hpf (*Figure 2A–D'*), similar to the above-described AgRP neurons. At both 36 hpf and 84 hpf, POMCa is also expressed in the pituitary gland, which is apposed onto to the hypothalamus midline and clearly recognizable at 84 hpf as a single structure with closely condensed cells (*Figure 2A–D'*). At 36 hpf, we count significantly more POMCa neurons in SF than in CF (*Figure 2E*; of note at this stage hypothalamic and pituitary neurons were very difficult to distinguish and were therefore pooled). At 84 hpf, SF display more POMCa neurons than CF in the pituitary (now well identifiable; *Figure 2F*) but not in the hypothalamus (*Figure 2E and V*).

POMCb is first expressed at 18 hpf at the rostral tip of the developing hypothalamus (*Figure 1—figure supplement 2*). At 24 hpf, a cluster of POMCb neurons is located at the dorsal limit of the hypothalamus basal plate, bordering the ORR (*Figure 2G–H'*). At 36 hpf, POMCb and POMCa are partially co-expressed in the anterior hypothalamus (*Figure 2I–J' and W*), with POMCb extending more dorsally close to the ORR, and POMCa more ventrally. At 84 hpf, POMCb neurons form two elongated stripes in the tuberal hypothalamus, but unlike POMCa, many POMCb neurons lie along the poc close to the hypothalamus/ORR border (*Figure 2K–L'*). Some of these POMCb neurons extend posteriorly, so that Hcrt and POMCb cells are very close to each other (compare *Figure 1M'-N'* and *Figure 2K'-L'*), but the two peptides are never co-expressed (*Figure 2W*). POMCb is also detected in the pituitary from 36 hpf onwards (*Figure 2I–L'*). From 36 hpf, POMCb neurons become more abundant in both the pituitary and the hypothalamus of SF, reaching about twice more cells at 84 hpf than in CF (*Figure 2M–N*).

Another neuropeptide expressed in this same region is AVT. It is expressed from 24 hpf in a very similar pattern to POMCb, and the two peptides indeed partially co-localize in some cells (*Figure 2O–T', W*). Between 36 hpf and 84 hpf, the number of AVT neurons in the hypothalamus keeps increasing in SF but not in CF so that AVT neurons are more abundant in SF (*Figure 2U*). In addition, two symmetrical clusters appear in the lateral and posterior ORR at 84 hpf (*Figure 2S–T'*), which we propose to belong to the teleost-specific NPO (*Herget et al., 2014*) (*Figure 1—figure*

**Figure 2.** Comparative development of POMCa, POMCb and AVT neurons in SF and CF. (A–T') Photographs of embryonic brains after in situ hybridization for POMCa, POMCb and AVT at 24, 36, and 84 hpf. The stages, the lateral or ventral orientations, and the probes are indicated. Blue squares indicate peptidergic clusters with higher numbers of neurons in SF. (E, F, M, N, U) Quantification and time-course of cell numbers in specific clusters. Mann-Whitney tests. (V) Anatomical interpretation of peptidergic patterns and time-course of appearance. POMCb (stars), POMCa (circles) and

*Figure 2 continued on next page*

*Figure 2 continued*

AVT (triangles) neurons are reported on schematic embryonic brains, in ventral or lateral views. A color code indicates higher numbers of neurons in CF (red) or in SF (blue), or equivalent numbers (grey). ac, anterior commissure; hyp, hypothalamus; NPO, neurosecretory preoptic nucleus; or, optic recess; orr, optic recess region; pit, pituitary; poc, post-optic commissure; tel, telencephalon; tub, tuberal hypothalamus; sv, *saccus vasculosus*. (W) Confocal pictures after double fluorescence in situ hybridization for POMCa/POMCb (left), POMCb/AVT (middle) and Hcrt/POMCb (right), showing colocalisation or lack of colocalisation of the indicated neuropeptides.
DOI: https://doi.org/10.7554/eLife.32808.010

The following figure supplements are available for figure 2:

**Figure supplement 1.** Comparative development of IT, CART and MCH neurons in SF and CF.
DOI: https://doi.org/10.7554/eLife.32808.011
**Figure supplement 2.** Comparative expression of *otpb* between 14 hpf and 36 hpf in SF and CF.
DOI: https://doi.org/10.7554/eLife.32808.012

*supplement 1C*). By contrast to the hypothalamus, the number of AVT cells in the NPO of SF and CF was not different (SF: 30.3 ± 2.7, n = 14; CF: 31 ± 3, n = 16).

## Similar numbers of IT, CART3 and MCH neurons in developing cavefish and surface fish

The three other neuropeptides we studied are CART3, IT, and MCH (*Figure 2—figure supplement 1*). The results are summarized in panel T and the raw cell counts are provided as *Supplementary file 1*. IT is expressed in two symmetrical groups in the lateral and posterior ORR, corresponding to the NPO as demonstrated by *Otpb* co-expression, and starting between 24 hpf and 36 hpf (*Figure 2-figure supplement 1A–E',U*). IT neurons counts were remarkably identical in the NPO of the two morphs at all stages examined (*Figure 2—figure supplement 1G,T*). CART3 is transiently expressed by more neurons in the SF telencephalon at 24 hpf and in the CF NPO at 36 hpf but none of these differences are maintained at 84 hpf (*Figure 2—figure supplement 1H–M',N,T*). Finally, MCH onset of expression is around 36 hpf, a stage where more MCH neurons are observed in a very discrete cell cluster of the posterior tuberal hypothalamus in CF (*Figure 2—figure supplement 1O–R',S*). At 84 hpf, the difference is not maintained, and a few MCH cells also appear in the mammillary region of the hypothalamus (*Figure 2—figure supplement 1Q–R',S,T*).

In sum, the comparative developmental maps of neuropeptidergic cell clusters presented above demonstrate region-specific, neuropeptide-specific, and time-specific differences in peptidergic neuron development in the two *Astyanax* morphs. There is a general trend for cavefish to possess more NPY, Hcrt, AgRP and less POMCa, POMCb, AVT neurons, even though these different types of neurons develop from the same or very close neuroepithelial zones in the hypothalamus basal plate, at the border of the ORR/PO region (examples: Hcrt/POMCb or AgRP/AVT). In line with this observation, these two categories of neuropeptides are never co-expressed. Interestingly, all peptidergic cell types showing quantitative variations in numbers between SF and CF were located in the basal plate of the hypothalamus. By contrast, neuropeptidergic cells located in the alar plate, in the NPO, displayed no difference between the two morphs. Finally, the data also show that in *Astyanax*, neuropeptidergic neurons show terminal differentiation signs (expression of their peptide transmitter) very early in embryogenesis, and therefore must be born equally early. The earliest expressed neuropeptides are Hcrt and POMCb (18hpf), followed by NPY, AVT, IT, CART3 (24hpf) and finally AgRP, POMCa, MCH (36hpf). The earliest differentiated neuropeptidergic cells are located in the dorsal hypothalamus at the frontier with the ORR, in a zone around the poc (*Figures 1U* and *2V*; *Figure 1—figure supplement 1*).

## Lhx7 and Lhx9 delineate NPY and Hcrt neuron territories and specify their peptidergic phenotypes

We next sought to define some of the causal molecular determinism for the above-described differences, taking advantage of the cavefish 'natural mutant' to pinpoint developmental mechanisms that can participate in the evolution of the hypothalamic networks in natural populations.

Previous analyses in zebrafish have defined the posterior half of the *otpa* (*orthopedia a*) domain as the larval NPO and as the homologous region to the mammalian PVN (*Herget et al., 2014*; *Herget and Ryu, 2015*). In *Astyanax* embryos, *otpb* (and *otpa*, not shown) were also strongly

expressed in the region of the ORR corresponding to the NPO, and indeed co-localized with IT (*Figure 2—figure supplement 2* and *Figure 2—figure supplement 1U*). In line with the lack of difference of peptidergic populations in the NPO between SF and CF embryos, there was no difference in *otpb* (or *otpa*, not shown) expression patterns between the two morphs, between 14 hpf and 36 hpf (*Figure 2—figure supplement 2*).

For neuronal populations which vary between the two morphs and are located in the hypothalamic basal plate close to the ORR, we used a candidate approach focusing on two LIM-homeodomain TFs. We first built a co-expression map between *Lhx7* and *Lhx9,* and NPY, Hcrt and POMCb, at 24 hpf. The results described below apply for both SF and CF, as we did not observe qualitative differences between the two morphotypes (*Figure 3* and *Figure 3—figure supplement 1*).

*Lhx7* and *Lhx9* are both expressed in the region of interest around the poc and they are not co-expressed (*Figure 3A*). *Lhx9* is seen in the hypothalamus, and *Lhx7* is expressed at the ORR/hypothalamic border (i.e. more dorsal than *Lhx9* according to the axis). In a lateral confocal plan, the two genes form diagonal domains parallel to the optic recess (*Figure 3A*). As neuronal differentiation occurs in a centrifuge fashion, from the progenitors lining the ventricular surface to the post-mitotic neurons at a distance from them in the mantle (Arrows in *Figure 1—figure supplement 1B*) (*Affaticati et al., 2015*), the *Lhx7* and *Lhx9* patterns suggest that the 2 TFs are expressed in post-mitotic neurons.

All Hcrt neurons express *Lhx9* (*Figure 3B*). *Lhx9+/Hcrt-* cells are also visible around the Hcrt population. To further assess the functional role of *Lhx9* in Hcrt specification, we performed *Lhx9* morpholino (MO) knock-down and counted Hcrt neurons at 24hpf. Three different MOs were used: AUG MO, 5'UTR MO, and splice blocking MO. The latter was controlled for effective *Lhx9* knock-down by RT-PCR (*Figure 3—figure supplement 2A*). All 3 *Lhx9* CF morphants types had reduced numbers of Hcrt neurons in the hypothalamus when compared to mismatch (mm) control MO (*Figure 3CD* and *Figure 3—figure supplement 3A*). Moreover, co-injection of *Lhx9* mRNA with the splice blocking MO rescued the phenotype (*Figure 3D*). Conversely, embryos injected with *Lhx9* mRNA had more Hcrt neurons (*Figure 3CD*). Thus, *Lhx9* has a conserved role in Hcrt neurons specification in *Astyanax* when compared to mouse or zebrafish (*Dalal et al., 2013*; *Liu et al., 2015*).

All NPY neurons express *Lhx7* (*Figure 3E,H*). At 24hpf and 84hpf (not shown), all NPY cells identified as the future *saccus vasculosus* express *Lhx7* - but only a subset of *Lhx7+* cells express *NPY* (*Figure 3E*). At 84 hpf, NPY cells located at the ORR/hypothalamic border also express *Lhx7* (*Figure 3H*). In addition, while some Hcrt neurons fall into the *Lhx7* domain and *Hcrt+* and *Lhx7 +* cells were clearly intermingled, we could not detect any cell co-expressing the two genes (*Figure 3I*). Thus, *Lhx7* is a good novel candidate to play a role in NPY-ergic specification, a hypothesis which was tested through morpholino knock-down. *Lhx7* morphant CF (either AUG MO or 5'UTR MO or splice blocking MO; *Figure 3—figure supplement 2B* for controls) had less NPY neurons in the presumptive *saccus vasculosus* at 24 hpf (*Figure 3FG* and *Figure 3—figure supplement 3B*). Co-injection of *Lhx7* mRNA with the splice blocking MO rescued the phenotype (*Figure 3G*). Conversely, embryos injected with *Lhx7* mRNA showed more NPY neurons (*Figure 3FG*). These data demonstrate a novel role for *Lhx7*, which is necessary and sufficient for NPY neurons specification.

In contrast to NPY and Hcrt neuropeptides, we could not find any expression of the two LIM-hd TFs in POMCb neurons. In fact, POMCb neurons are located within the *Lhx9* and *Lhx7* domains, but they are intermingled in a salt-and-paper fashion among cells expressing the two TFs (*Figure 3JK*). POMCb cells are also excluded from the superficial acroterminal *Lhx7* domain (*Figure 3K*). In summary, POMCb neurons (less abundant in CF) are surrounded by NPY and Hcrt neurons (more abundant in CF) that express distinct TFs. In line with these expression data, the number of POMCb cells at 24 hpf was unaffected after *Lhx9* or *Lhx7* MO knock-down (*Figure 3—figure supplement 4A–C*). These results also confirm the specificity of Lhx9 and Lhx7 TFs functions toward Hcrt and NPY fates, respectively. These data are summarized in the schematic *Figure 3L*.

Finally, we noticed that although non-statistically significant, POMCb cell numbers had a tendency to be slightly higher in CF *Lhx7* morphants than in their controls (*Figure 3—figure supplement 4C*). Given the anatomical organization of POMCb cells relatively to *Lhx9*/Hcrt cells and *Lhx7*/NPY cells, we next tested the possibility of a balance between POMCb versus Hcrt/NPY specification at progenitor level by double *Lhx9/Lhx7* knock-down in CF. However, the number of POMCb cells were unchanged in the double morphants (*Figure 3—figure supplement 4D*).

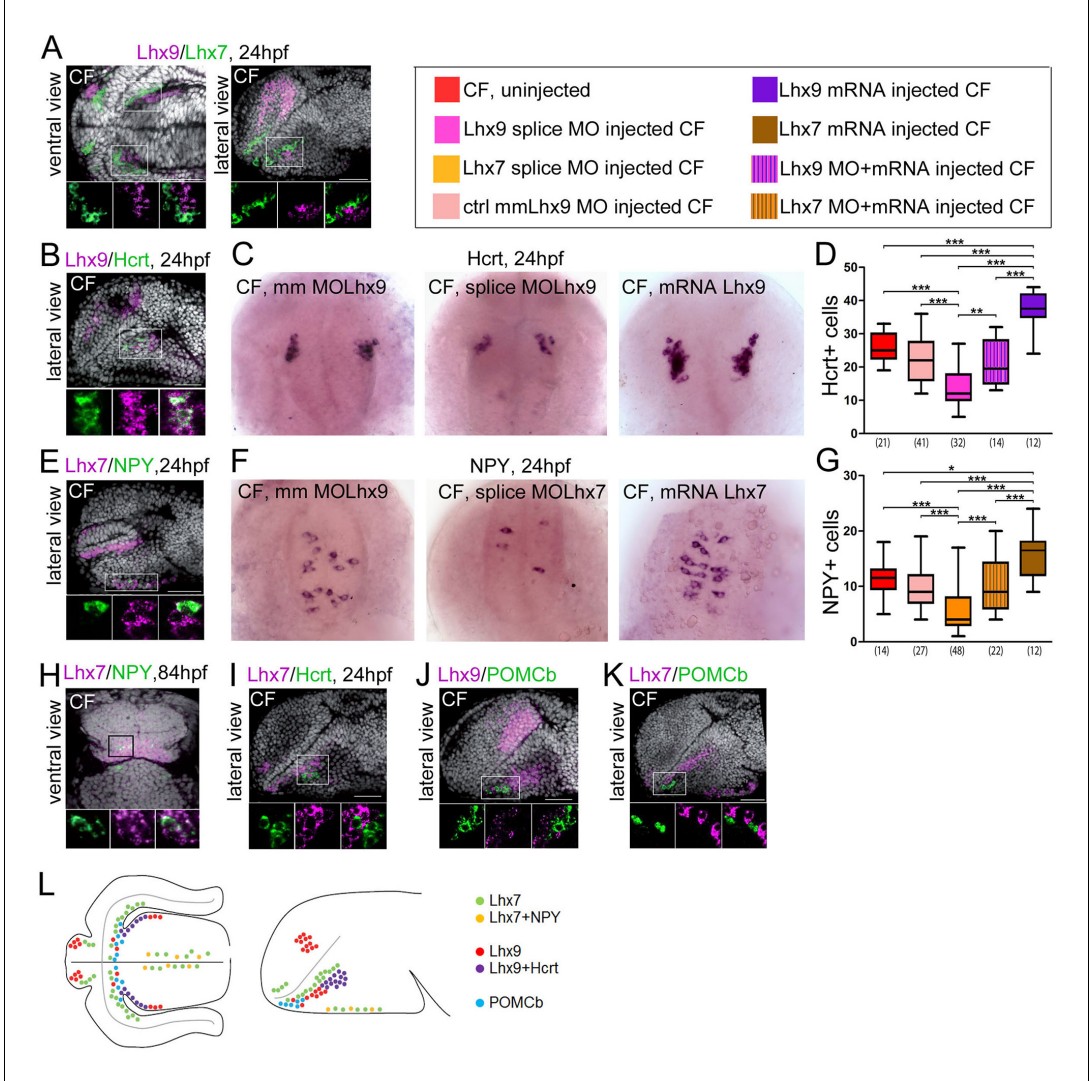

**Figure 3.** Expression and roles of Lhx7 and Lhx9 in NPY and Hypocretin neurons development. (A, B, E and H–K) Confocal pictures after double fluorescence in situ hybridization in CF (one probe in magenta, the other in green, as indicated) with DAPI counterstain (grey nuclei). On all panels, the orientation, the stage, and the probes are indicated. The top photos show low magnification pictures of the whole forebrain, and the bottom photos show high power views for assessment of co-localization. (C) Photographs of control mismatch (mm), *Lhx9* e1-i1 splice MO and *Lhx9* mRNA-injected CF embryos after Hcrt in situ hybridization at 24 hpf. (D) Quantification of the number of Hcrt cells in control mismatch (pale red, 0.48 mM), *Lhx9* splice MO-injected (pink, 0.48 mM), *Lhx9* mRNA (purple, 200 ng/µl) or *Lhx9* splice MO + *Lhx9* mRNA (striped) injected CF. Un-injected CF are in red. See color code. ANOVA tests. In this and the following figures, numbers under boxplots indicate the numbers of embryos examined. (H) Photographs of control mismatch (mm), *Lhx7* e4-i4 splice MO and *Lhx7* mRNA-injected CF embryos after NPY in situ hybridization at 24hpf. (I,) Quantification of the number of NPY cells in control mismatch (pale red, 0.96 mM), *Lhx7* splice MO-injected (orange, 0.96 mM), *Lhx7* mRNA (brown, 200 ng/µl) or *Lhx9* splice MO + *Lhx7* mRNA (striped) injected CF. Un-injected CF are in red. See color code. ANOVA tests. (L) Summary of Lhx7/Lhx9/NPY/Hcrt/POMCb expression. A color code indicates the presence or absence of co-localization.

DOI: https://doi.org/10.7554/eLife.32808.013

The following figure supplements are available for figure 3:

**Figure supplement 1.** Expression of *Lhx7*, *Lhx9*, *Hcrt* and *NPY* in SF.

DOI: https://doi.org/10.7554/eLife.32808.014

**Figure supplement 2.** Controlling *Lhx9* and *Lhx7* splice-blocking morpholinos knock-down efficiency.

DOI: https://doi.org/10.7554/eLife.32808.015

**Figure supplement 3.** Effects of *Lhx9* or *Lhx7* ATG and 5'UTR morpholinos knock-down on Hcrt and NPY neurons numbers, respectively.

DOI: https://doi.org/10.7554/eLife.32808.016

**Figure supplement 4.** Effects of *Lhx9* or/and *Lhx7* morpholino knock-down on POMCb neurons numbers.

DOI: https://doi.org/10.7554/eLife.32808.017

## Expansion of Lhx7 and Lhx9 domains precedes the onset of Hcrt and NPY expression

We next reasoned that if *Lhx7* and *Lhx9* are involved in NPY and Hcrt specification, respectively, then their expression should precede neuropeptide expression, and may be spatially or temporally different between CF and SF embryos.

At 14-15 hpf, 4 hrs prior to the onset of Hcrt expression, *Lhx9* is expressed in the optic vesicles, but also in a stripe of cells adjacent to it (*Figure 4A*). This latter *Lhx9* domain topologically corresponds to the future Hcrt expression site. All CF but not all SF embryos present this stripe of cells at 14 hpf, suggesting that in this domain *Lhx9* expression is turned on slightly earlier in CF than in SF (*Figure 4—figure supplement 1*). We have then quantified the length of the *Lhx9* domain at 15 hpf and found that it is expanded in CF (*Figure 4A*). Indeed, while in CF these *Lhx9*+ cells extend all along the entire optic vesicle in SF they are restricted to the anterior half of the optic vesicle.

At 22 hpf, 2 hr prior to the onset of *NPY* expression in the acroterminal floor, *Lhx7* expression is already similar to that described above at 24 hpf (*Figure 4B*). Its expression is expanded in the superficial floor of the forming hypothalamus in CF (*Figure 4B*), where we found a higher number of NPY neurons 2 hr later as compared to SF. In ventral view the area covered by *Lhx7* +cells and the number of *Lhx7* +cells is higher in CF than SF and in lateral view the *Lhx7* +domain is extended in CF (*Figure 4B*). Moreover, in the dorsal hypothalamus, where we observed an increased number of NPY neurons at 84hpf in CF, we found an expanded *Lhx7* expression spot at 60 hpf, that is, 24 hr before the appearance of these neurons (*Figure 4C*), and in line with previous findings from the group (*Menuet et al., 2007*).

Taken together, these results show that the increased numbers of differentiating NPY and Hcrt neurons in the CF basal forebrain are preceded by a parallel expansion of their co-expressed *Lhx* genes. Although the functional analyses and the spatio-temporal co-expression data we provide cannot strictly be taken as an indication on the lineage of NPY and Hcrt neurons, they strongly suggest that the differences in hypothalamic peptidergic neuron patterning between SF and CF originate in the early embryonic neuroepithelium (e.g. before 14 hpf for *Lhx9*/Hcrt). Thus, we next decided to investigate whether the changes in neuropeptidergic cell numbers observed in CF are the consequence of the same embryonic signaling events that affect its neural tube and eye morphogenesis.

## Fgf and Shh signaling during gastrulation control the future numbers of specific peptidergic neuronal populations

We reasoned that neuronal populations which vary between the two morphs are located in the basal hypothalamus, so that their progenitors are subjected to the influence of ventral midline, prechordal plate and basal plate Shh signaling, as well as anterior neural ridge/acroterminal domain Fgf signaling. In cavefish, heterotopic (larger) expression of *Shh* at the ventral midline and heterochronic (earlier) expression of *Fgf8* at the ANR impact the anterior neural plate fate map and lead to morphogenetic defects (*Yamamoto et al., 2004*; *Pottin et al., 2011*). We hypothesized that in cavefish, the increased numbers of NPY and Hcrt neurons and the decreased numbers of POMCb neurons may result from changes in above signaling systems in the early neuroepithelium.

### NPY neurons

To test this hypothesis, we performed cyclopamine treatments (50 µM; Shh signaling inhibitor) between 8 and 12 hpf in CF to mimic the SF condition. This resulted in a massive reduction of *Lhx7* expression in the prosencephalon, but without affecting at all the *Lhx7* acroterminal/putative *saccus vasculosus* domain at 24hpf (*Figure 5AB*), showing that the acroterminal and the hypothalamic NPY cell groups are controlled through distinct signaling mechanisms. Accordingly, the same cyclopamine treatment did not affect the number of NPY cells in the acroterminal region at 24 hpf (*Figure 5B*). As NPY hypothalamic cells arise later, around 84 hpf (*Figure 1*), and cyclopamine treatments affect later larval survival, we did not assess cyclopamine effect on 84 hpf larvae. We used instead a treatment with the FgfR signaling inhibitor SU5402 (5 µM; between 8 hpf and 12 hpf), which results in a decrease of *Shh* expression in CF, therefore also indirectly mimicking the SF phenotype (*Hinaux et al., 2016*; *Pottin et al., 2011*), but with less deleterious effects on larval development. In line with the effect of cyclopamine on the hypothalamic *Lhx7* domain at 24 hpf, and with the role of *Lhx7* in NPY specification shown above, SU5402 treatment on CF caused a reduction of

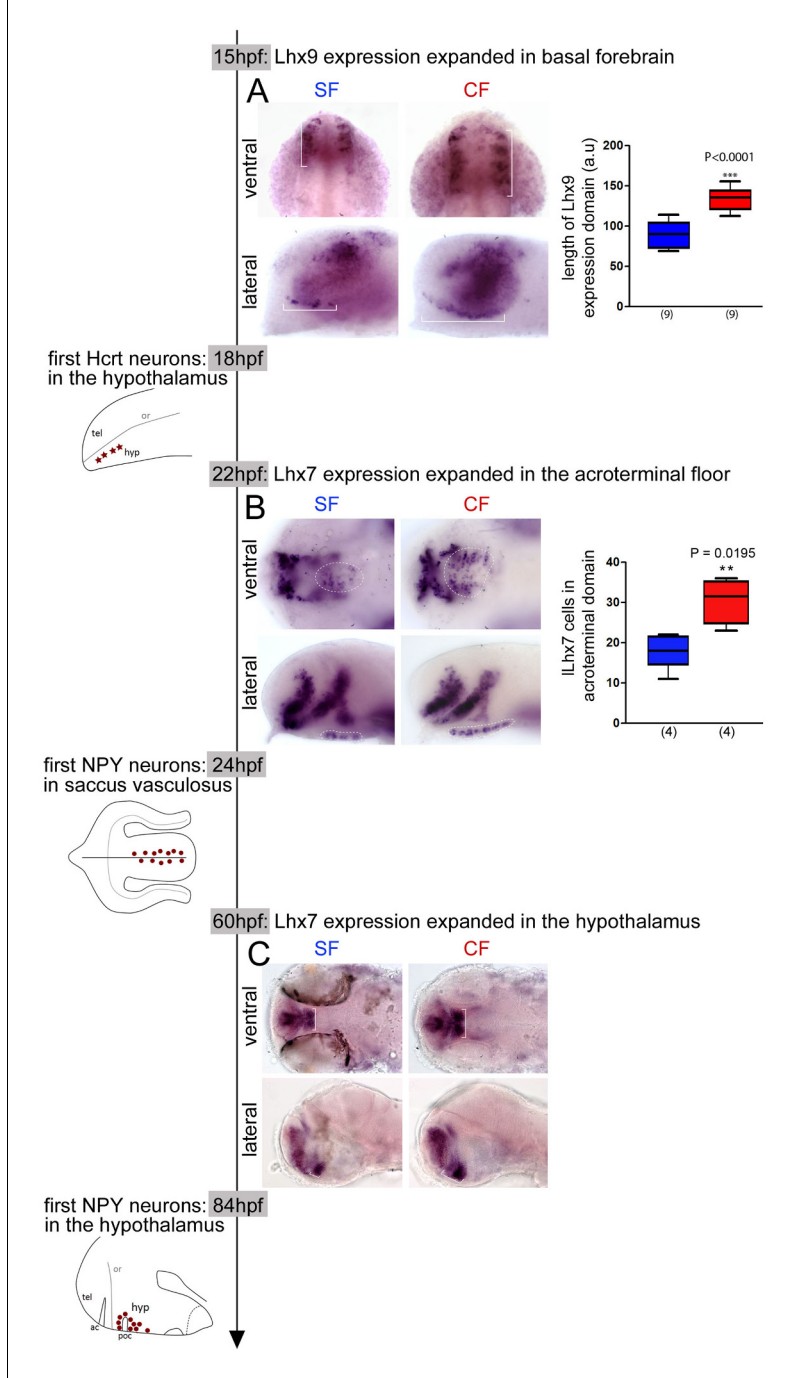

**Figure 4.** Time-line of expression of *Lhx9* and *Lhx7* relative to Hcrt and NPY neuron differentiation between 15 hpf and 84 hpf in SF and CF. (**A**) Photographs of embryonic brains after in situ hybridization for *Lhx9* at 15 hpf in SF and CF, in lateral and ventral views, and quantification of the hypothalamic expression domain (in brackets on the pictures). SF in blue, CF in red. (**B**) Photographs of embryonic brains after in situ hybridization for *Lhx7* at 22 hpf in SF and CF, in lateral and ventral views, and quantification of the acroterminal expression domain (in dotted lines on the pictures). SF in blue, CF in red. (**C**) Photographs of larval brains after in situ hybridization for *Lhx7* at 60 hpf in SF and CF, in lateral and ventral views.

DOI: https://doi.org/10.7554/eLife.32808.018

The following figure supplement is available for figure 4:

**Figure supplement 1.** A slight heterochrony in the onset of *Lhx9* expression between SF and CF.

DOI: https://doi.org/10.7554/eLife.32808.019

hypothalamic NPY cell numbers at 84 hpf (*Figure 5C*). Conversely, SU5402 had no effect on *Lhx7* expression or NPY neuron numbers in the acroterminal domain at 24 hpf or 84 hpf, confirming that Shh and Fgf signaling are not involved in the control of the size of this group of NPY cells (*Figure 5D*). As the acroterminal domain is a source of Fgf signaling according to the prosomeric model (*Puelles and Rubenstein, 2015*), and is also very close to Shh signaling sources in the hypothalamic basal plate, we sought to test the possibility that at later stages these signaling systems could affect the development of the future *saccus vasculosus* NPY cells. To this aim, CF embryos were treated with cyclopamine or SU5402 between 20 and 24 hpf, that is, during the time window when these NPY cells differentiate. However, neither of these treatments changed the number of acroterminal NPY cells assessed at 24 hpf (*Figure 5—figure supplement 1A*).

### Hcrt neurons

Neither cyclopamine (50 μM, 8-12 hpf) nor SU5402 (5 μM, 8 hpf-12 hpf) treatment affected *Lhx9* expression at 15 hpf (*Figure 5EF*). Cyclopamine did not change the numbers of Hcrt neurons at 24 hpf either (*Figure 5G*) suggesting that Hcrt progenitors are not under the control of these signaling systems. We considered this result surprising as Hcrt cells (contrarily to NPY cells; *Figure 5—figure supplement 1B*) clearly belong to the basal hypothalamus *Shh+* lineage (*Figure 5H*; see also [*Alvarez-Bolado et al., 2012*]). We thus performed the cyclopamine treatment slightly later, between 16 and 20 hpf. Such a treatment done during the differentiation of the first Hcrt neurons resulted in a decrease in Hcrt neurons numbers (*Figure 5IJ*), suggesting a positive control by Shh signaling on Hcrt differentiation (but not early specification). Of note, the same type of treatment with cyclopamine (16-20 hpf) or SU (16-20 hpf) did not affect POMCb cell numbers (*Figure 5—figure supplement 1C*), demonstrating further the specificity of Shh and FgfR signaling toward neuropeptidergic neurons development. Surprisingly, SU5402 (5 μM, 8 hpf-12 hpf) treatment on the other hand systematically resulted in a reduction of Hcrt cell numbers at 18 hpf or 36 hpf, persisting at 84 hpf (*Figure 5K*). This suggests that in this case, (1) the effect of early FgfR signaling is not mediated through a reduction of Shh signaling – as early cyclopamine treatment does not affect Hcrt cell numbers, and (2) FgfR signaling may be necessary or permissive to allow Lhx9 TF effect on Hcrt fate – as SU5402 diminishes Hcrt but not Lhx9 cell numbers.

In summary, this series of experiments provide evidence that Shh and FgfR signaling at the end of gastrulation and during neurulation influence later hypothalamic neuroanatomy, and thus are, at least in part, responsible for the developmental variation in neuropeptidergic patterning in the cavefish brain.

## Behavioral consequences of the developmental evolution of neuropeptidergic patterning in cavefish

Finally, we sought to assess the functional outcomes of the above described variations in peptidergic neurons development in CF.

There is a striking general trend for cavefish to possess more orexigenic NPY, Hcrt and AgRP neurons (which stimulate food intake) and less anorexigenic POMCa, POMCb and VP neurons (which inhibit food intake; as shown above these two categories of neuropeptides are never co-expressed). Therefore, we first compared food intake between CF and SF larvae. We set up a test in which individual larvae were given an excess of *Artemia* nauplii, and the number of *Artemia* eaten over a period of 5 hr were counted. The test was performed on larval stages from 5.5 dpf (days post-fertilization, i.e. after mouth opening) to 26 dpf. At all ages studied, CF and SF larvae eat consistently the same amounts of food (*Figure 6A*). The results were identical after normalization to the size of the larvae, which may slightly vary between individuals after 10 dpf (not shown). This result is in line with recently published data showing that adult Pachón cavefish and their surface conspecifics have a comparable appetite (*Aspiras et al., 2015*), and further shows that larval food intake is also identical between the two morphs.

This result prompted us to examine alternative potential consequences for CF of having more of certain neuropeptidergic types. We focused on Hcrt, because this neuropeptide is also well known for its role in locomotion and sleep regulation, and CF larvae and adults show a dramatic change in these traits when compared to SF (*Yoshizawa et al., 2015*; *Duboué et al., 2011*). We thus compared locomotor activity in 1-week-old *Lhx9* morphants, their controls, and SF. Importantly, 7 dpf

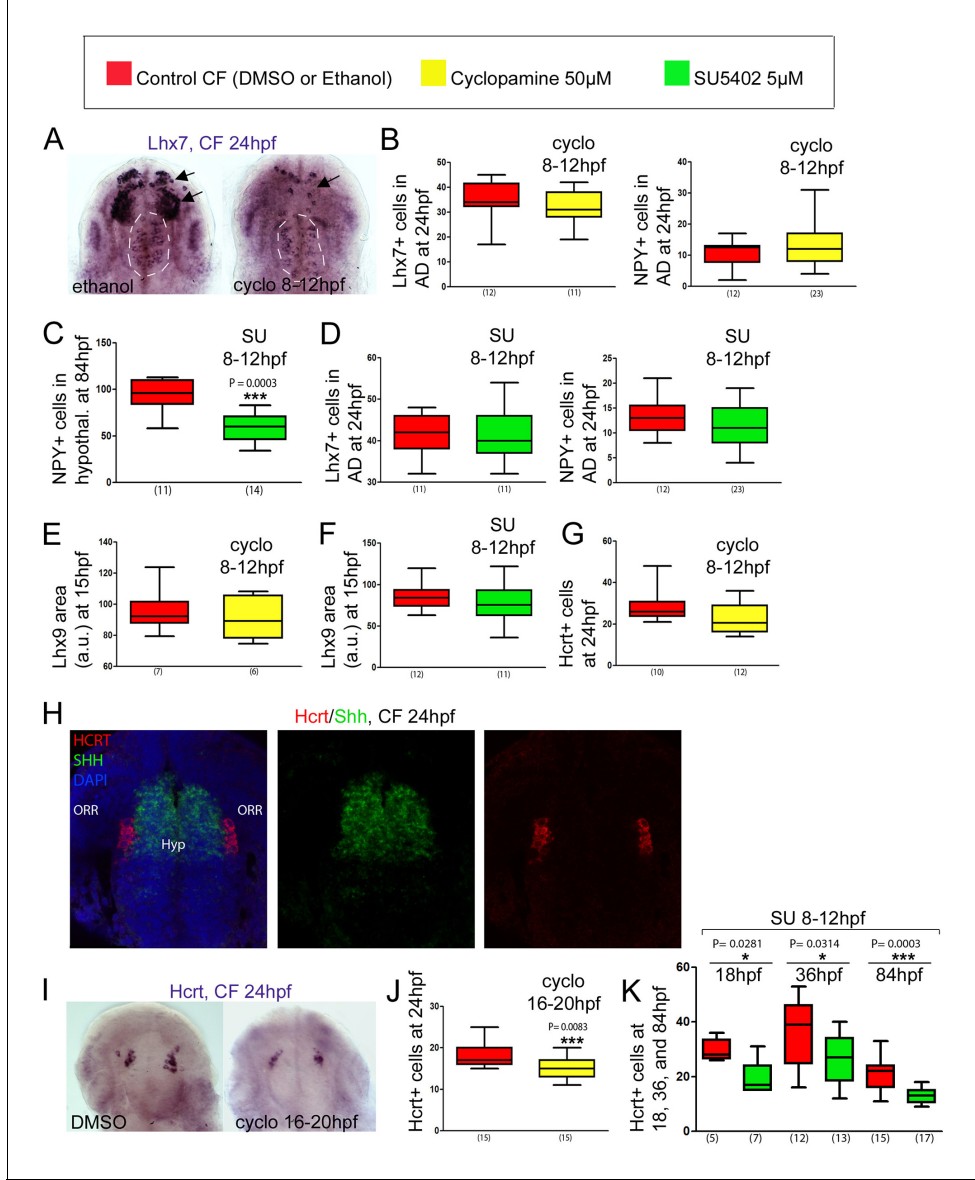

**Figure 5.** Effect of early or late inhibition of Shh and Fgf signaling on Lhx7, Lhx9, NPY, Hcrt and POMCb expression. (**A**) Photographs of embryonic brains after in situ hybridization for *Lhx7* at 24 hpf in control and cyclopamine-treated CF, in ventral views. The treatment was performed between 8 and 12 hpf. The presumptive *saccus vasculosus* (unaffected by the treatment) is delineated by a white dotted line, whereas the other, affected, expression domains are indicated by black arrows. (**B**) Quantification of the number of Lhx7 cells (left) and NPY cells (right) in the acroterminal domain (AD) of control CF (red) and cyclopamine-treated CF (yellow) at 24 hpf after an 8-12 hpf treatment. (**C**) Quantification of the number of NPY cells in the hypothalamus of control CF (red) and SU-treated CF (green) at 84 hpf after an 8-12 hpf treatment. Mann-Whitney tests. (**D**) Quantification of the number of Lhx7 cells (left) and NPY cells (right) in the AD of control CF (red) and SU-treated CF (green) at 24 hpf after a 8-12 hpf treatment. (**E, F**) Quantification of the Lhx9-expressing area in the hypothalamus of control CF (red) versus cyclopamine-treated (yellow)(**E**) or SU-treated CF (green)(**F**) at 15 hpf, after an 8-12 hpf treatment. (**G**) Quantification of the number of Hcrt cells in the hypothalamus of control CF (red) and cyclopamine-treated CF (yellow) at 24 hpf after an 8-12 hpf treatment. (**H**) Confocal pictures after double fluorescence in situ hybridization (Hcrt in red, Shh in green, as indicated) with DAPI counterstain (blue nuclei) at 24 hpf, in ventral view. (**I**) Photographs of embryonic brains after in situ hybridization for *Hcrt* at 24 hpf in control and cyclopamine-treated CF, in ventral views. The treatment was performed between 16 and 20 hpf. (**J**) Quantification of the number of Hcrt cells of control CF (red) and cyclopamine-treated CF (yellow) at 24 hpf after a 16-20 hpf treatment. Mann-Whitney tests. (**K**) Quantification of the number of Hcrt cells of control CF (red) and SU-treated CF (green) at 18, 36, and 84 hpf, after an 8-12 hpf treatment. Mann-Whitney tests.

DOI: https://doi.org/10.7554/eLife.32808.020

The following figure supplement is available for figure 5:

**Figure supplement 1.** Late inhibition of Shh or Fgf signaling has no effect on acroterminal NPY or POMCb differentiation.

DOI: https://doi.org/10.7554/eLife.32808.021

*Lhx9* CF morphants still had less Hcrt neurons in their hypothalamus than controls, thereby mimicking the SF situation (*Figure 6B*), and showing that the effect of *Lhx9* knock-down has not been compensated. Behavioral recordings were then performed on 24 hr cycles. Seven dpf CF larvae are about 30–50% more active than SF (*Figure 6CD*), hence the difference in locomotor activity in the 7 dpf larvae of the two morphs is similar to that observed in older fish (*Yoshizawa et al., 2015*; *Duboué et al., 2011*) or younger fish (*Pottin et al., 2010*). Of note, the locomotion did not vary significantly according to the day/night periods, although some level of diurnal rhythms has been reported in 3-week-old fish and in adults (*Yoshizawa et al., 2015*; *Duboué et al., 2011*). This may relate to the developmental establishment of rhythmicity. Importantly, the distances swam by the 7 dpf *Lhx9* morphant CF with reduced numbers of Hcrt neurons was identical to SF, both during day and night (*Figure 6CD* and *Figure 6—figure supplement 1A*), suggesting a role for the CF supernumerary Hcrt neurons in the control of this behavior. *Lhx9* CF morphants behave like SF, while mm*Lhx9* MO-injected CF behave like un-injected CF (*Figure 6D*), showing that the injection by itself does not affect swimming behavior. We also quantified the time spent by the larvae in a state of low activity (*Kalueff et al., 2013*; *Mathuru et al., 2012*). SF and *Lhx9* CF morphants spent 2 and 2.2 more time, respectively, in a hypoactive state than control CF (*Figure 6—figure supplement 1BC*). Therefore, the developmental control of hypothalamic peptidergic circuits that we have uncovered in CF has important behavioral consequences.

## Discussion

In this paper, we provide evidence for developmental evolution of hypothalamic neuropeptidergic clusters that accompany adaptation of cavefish to life in the dark. We uncover some of the underlying developmental mechanisms, and we pinpoint behavioral consequences of such anatomical developmental variations. In our interpretations, we always consider SF as the 'wildtype controls' and CF as the 'natural mutants'.

### Comparative developmental neuroanatomy

The anatomical interpretation of the results was crucial to draw hypotheses on the causal mechanisms of neuropeptidergic variations between the two *Astyanax* morphs. In particular, reasoning in terms of brain axes in the frame of the updated prosomeric model (*Puelles and Rubenstein, 2015*), and taking into account the particularities of the teleost brain (*Herget et al., 2014*; *Affaticati et al., 2015*) was instrumental to understand that the most affected peptidergic populations in cavefish were located in the hypothalamic basal plate, while alar plate populations including those of the NPO were unchanged. In fact, the developmental origin of the neuropeptides we have studied is very similar in *Astyanax* fish and in mouse (*Díaz et al., 2014*) in terms of D/V and A/P coordinates (*Figure 7A*). For example, NPY, Hcrt, POMC and AgRP have basal progenitor sources while IT comes from alar sources and CART have alar +basal sources in both species. The only difference we observed in this regard is a double alar +basal origin of AVT cells in fish. The A/P organization seems also conserved with for example the fish Hcrt cluster in a posterior position that corresponds well to its origin in the peduncular hypothalamo-telencephalic prosomere (Hp1, tuberal part) in the mouse, or the AgRP fish cluster located in an anterior position that fits well with its acroterminal origin (tuberal part, arcuate nucleus) in the mouse (*Díaz et al., 2014*). We propose that early born fish NPY cells also belong to the acroterminal domain. In mouse, a population of NPY cells born in the tuberal (dorsal) part of the acroterminal domain migrates tangentially and ventrally inside the acroterminal domain to populate the arcuate nucleus (*Díaz et al., 2014*). In fish, acroterminal NPY cells also seem to migrate ventrally to end up in the position of the fish-specific *saccus vasculosus*, a circumventricular organ with poorly studied neurosecretory functions and that may serve as a sensor of seasonal changes in day length (*Castro et al., 1999*; *Nakane et al., 2013*; *Tsuneki, 1986*). These acroterminal NPY cells of fishes and mammals thus have a shared developmental origin but distinct final fates and functions. Finally and interestingly, early-born acroterminal and later-born hypothalamic NPY cells behaved strikingly differently with regards to their response to signaling influences emanating from the borders of the neural plate and tube, further emphasizing the importance of interpreting anatomical data according to models of brain development.

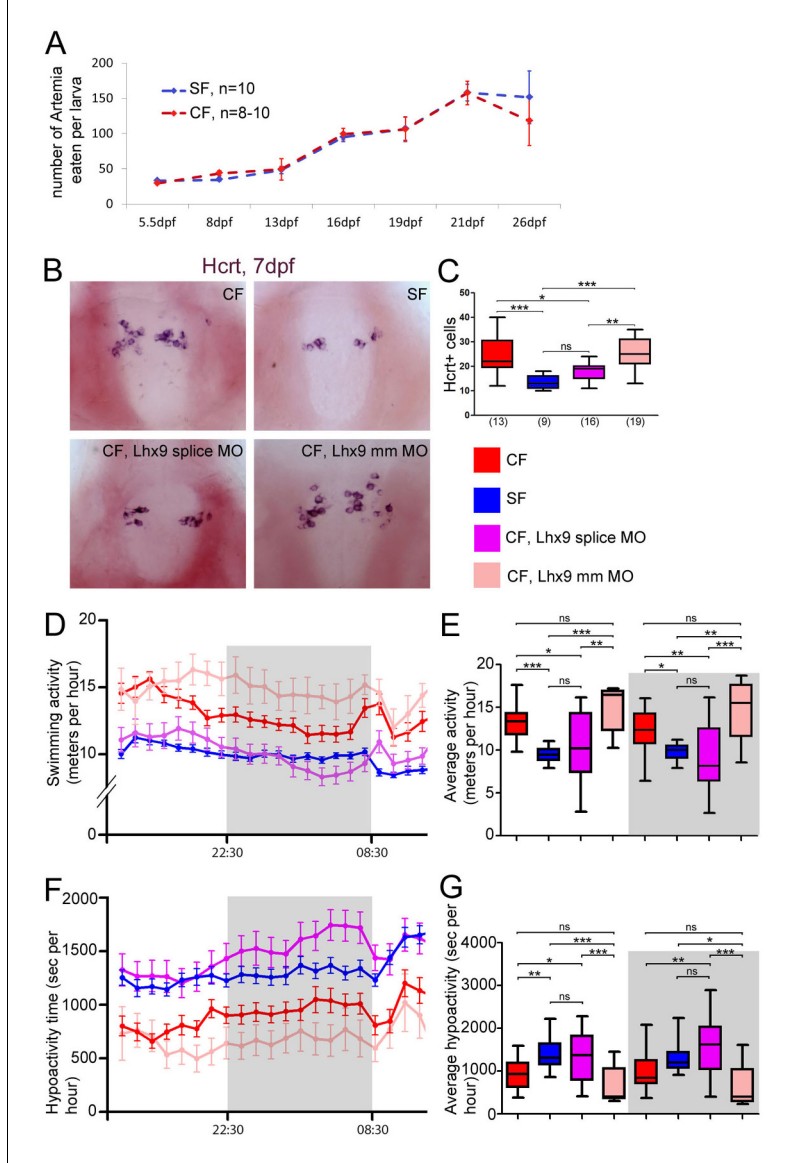

**Figure 6.** Neuropeptidergic evolution and cavefish larval behavior. (A) Comparison of food intake during larval and juvenile stages in SF (blue) and CF (red). (B, C) Photographs of brains after in situ hybridization for Hcrt, in ventral views, in 7 dpf juveniles, in the indicated condition (B) and quantification of Hcrt cell numbers at 7 dpf in CF (red), SF (blue), *Lhx9* splice MO-injected (purple) and mm*Lhx9* injected CF (pale red). (C). ANOVA test. (D, E) 24 hr plots of locomotor activity, and histogram showing average activity in the four conditions (see color code). The dark/night time is shaded in grey. In these and following graphs, the numbers of tested larvae are: CF, n = 23; SF, n = 21; *Lhx9* splice MO-injected CF, n = 33; mm*Lhx9* injected CF, n = 11. ANOVA test. (F, G) 24 hr plots of time spent in hypoactivity, and histogram showing average time spent in hypoactivity in the four conditions (see color code). The dark/night time is shaded in grey. ANOVA test.

DOI: https://doi.org/10.7554/eLife.32808.022

The following figure supplement is available for figure 6:

**Figure supplement 1.** Statistics on locomotion data on 24 hr.
DOI: https://doi.org/10.7554/eLife.32808.023

## Factors influencing the development of neuropeptidergic clusters

Neuropeptidergic clusters showing cell number differences between SF and CF are all located in the dorsal part of the hypothalamic basal plate, in a region topologically corresponding to the tuberal hypothalamus in mouse. Conversely, NPO clusters do not vary between the two morphs. Even more so, in the case of AVT which displays alar and basal clusters in *Astyanax*, the alar NPO cluster is

identical in SF and CF but the basal hypothalamic cluster is larger in SF. This strongly suggests that the neuroepithelial region giving rise to the 'tuberal' hypothalamus in CF is specifically subjected to patterning variations. Interestingly, in their new interpretation of prosencephalic development, Puelles et al. ascribe the induction of the hypothalamic basal plate in its ventrodorsal axis to the action of the migrating prechordal plate (*Puelles and Rubenstein, 2015*; *García-Calero et al., 2008*), rather than to the notochord. In cavefish, there is evidence for stronger Shh and dynamically modified Bmp4 signaling from the prechordal plate at the end of gastrulation (*Hinaux et al., 2016*; *Yamamoto et al., 2004*). On the other hand, Fgf8 secreted from the ANR potentially influences neuroepithelial regions topographically related to the anterior commissure, the rostral-most point of the neural tube, and also interacts with the Shh-secreting prechordal plate (*Hinaux et al., 2016*; *Pottin et al., 2011*). There, Fgf8 is expressed earlier in CF than in SF. Moreover, Fgf8/10/18 are also expressed at later stages in the acroterminal domain itself from embryonic day 11.5 in the mouse (*Ferran et al., 2015*) and from 15 hpf in zebrafish ([*Sbrogna et al., 2003*] and ZFIN), representing another potential source of signaling molecules.

Focusing on NPY, Hcrt and POMCb neurons, and using early (8-12 hpf) or later (16-20 hpf) pharmacological manipulations, we found that early Shh and FgfR signaling at neural plate stage positively regulate the size of a Lhx7/NPY cluster which differentiates in the hypothalamus at 84 hpf (*Figure 7B*). Conversely, the Lhx7/NPY cells of the acroterminal *saccus vasculosus* are insensitive to these signaling systems, and more investigations are needed to understand why they are more numerous in CF. Concerning the Lhx9/Hcrt cells, while an unknown mechanism must be responsible for the initial enlargement of the *Lhx9* expression domain at early stage, we uncovered a positive role of Shh/FgfR signaling during their differentiation but not their initial specification at neural plate stage, together with a putative 'priming' role of FgfR signaling for future Hcrt development (*Figure 7B*). Taken together, these data show that signaling variations at the end of gastrulation (i.e. unexpectedly very early) and during neurulation influence later hypothalamic progenitor dynamics and/or neuronal differentiation, and thus are, at least in part, responsible for the developmental variation in neuropeptidergic patterning in the cavefish brain. These data also add to the emerging knowledge on the role of signaling molecules on the successive steps of development of hypothalamic cell types (examples: [*Bosco et al., 2013*; *Muthu et al., 2016*; *Peng et al., 2012*; *Russek-Blum et al., 2008*]).

In addition, we provide evidence for a novel role of Lhx7, necessary and sufficient for the specification of NPY neurons. In the past years, several factors controlling specification and differentiation within specific nuclei or regions of the developing hypothalamus have been found (for review [*Bedont et al., 2015*]), among which LIM-homeodomain TFs are well represented. Indeed, Lhx9 is necessary and sufficient for Hcrt neuron specification in zebrafish and mouse (*Dalal et al., 2013*; *Liu et al., 2015*), a role that we also find conserved in *Astyanax*. Islet1 controls hypothalamic (but not pituitary) POMC/POMCa development in mouse and zebrafish, respectively (*Nasif et al., 2015*), and Lhx1 controls differentiation of several peptidergic types including VIP or Enk in the suprachiasmatic nucleus (*Bedont et al., 2014*). The present finding of the Lhx7/NPY link suggests the possibility of an emerging 'LIM code' for hypothalamic cell type development (*Bachy et al., 2002*; *Shirasaki and Pfaff, 2002*).

## From developmental evolution of neuronal clusters to behavior

The developing hypothalamus is larger in *Astyanax* CF than SF (*Menuet et al., 2007*; *Rétaux et al., 2008*). We reasoned that if all the hypothalamic neuronal populations of *Astyanax* CF were homothetically changed, then the net resulting behavioral and metabolic consequences should not be modified. But it is not the case. Some peptidergic clusters are larger, some are smaller, and some are unchanged. This prompted us to investigate potential functional behavioral consequences of these variations.

Within the arcuate nucleus of the mouse hypothalamus, neurons directly act on food intake and energy balance, with sometimes antagonist effects. Neurons that co-express NPY and AgRP, together with Orexin/Hcrt neurons, act to increase appetite (orexigenic). On the contrary, neurons that co-express CART and POMC decrease appetite (anorexigenic) (for reviews [*Loh et al., 2015*; *Larsen and Hunter, 2006*; *Millington, 2007*]). These systems are functionally conserved in fish (*Cerdá-Reverter et al., 2011*; *Volkoff and Peter, 2006*), including *Astyanax* cavefish (*Penney and Volkoff, 2014*; *Wall and Volkoff, 2013*). Here, we found that increased numbers of orexinergic

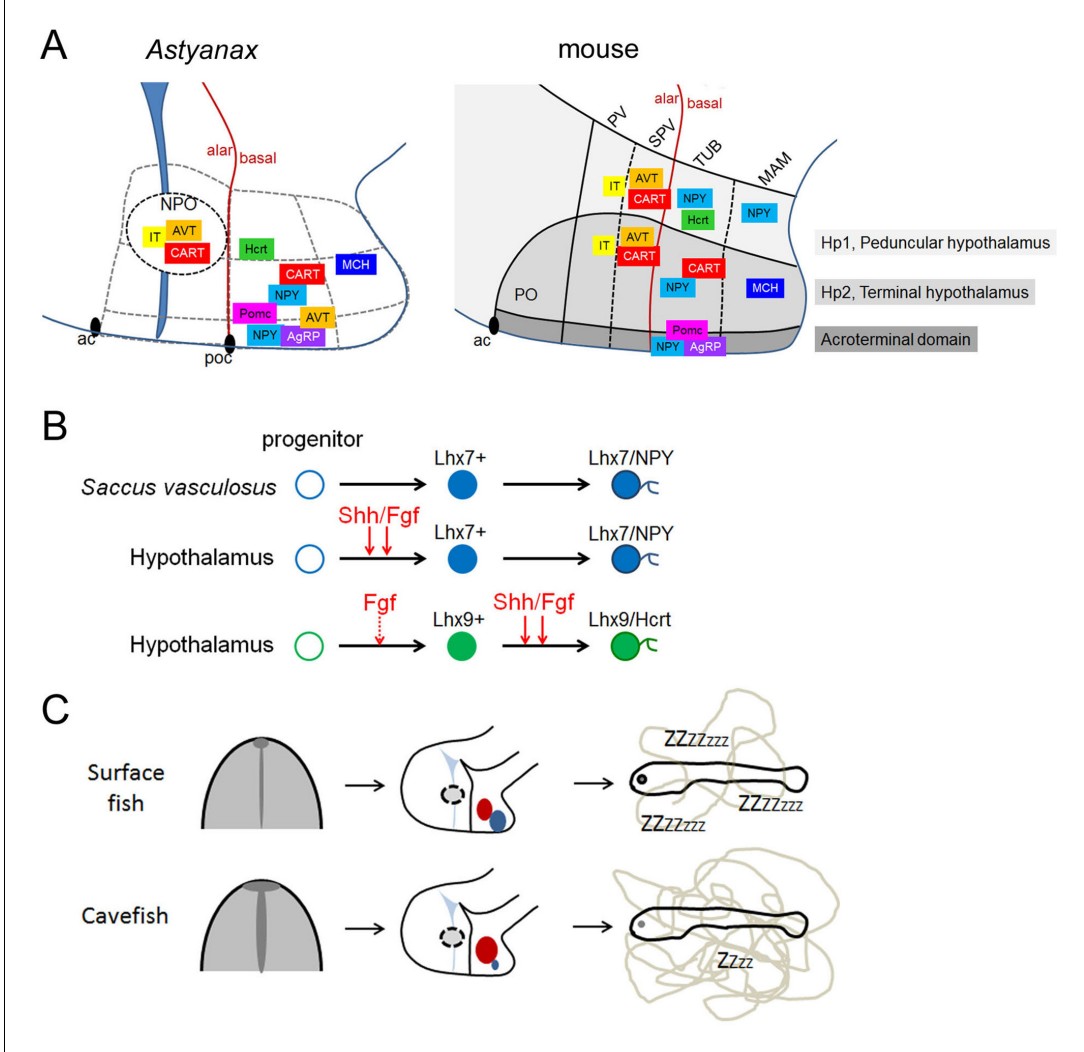

**Figure 7.** Summary schemes. (**A**) Comparative developmental maps of neuropeptidergic cell types between *Astyanax* and mouse. For mouse, data were taken from (*Díaz et al., 2014*). For *Astyanax mexicanus*, boundaries between putative hypothalamic subdivisions are grey dotted lines because we did not assess the position of the neuropeptidergic clusters within molecularly distinct cytoarchitectonic domains as it was done in the mouse (*Ferran et al., 2015*). Their relative positions are indicated, according to the brain axes and to the double labeling we have performed. Their localization in the mouse-like subdivisions is only tentative. (**B**) Signaling and transcription factors involved in NPY and Hcrt neurons development. Shh/Fgf signaling acts at different steps for the two neuropeptidergic types of neurons. The 'priming' effect of Fgf signaling on Hcrt fate without affecting *Lhx9* expression is indicated by a dotted line. (**C**) From early developmental evolution to behavioral consequences in cavefish. The SF/CF comparison is schematized at neural plate stage (left, with modified midline signaling in CF), at embryonic stage (middle, with modified hypothalamic peptidergic cluster sizes in CF), and in terms of larval behavior (right, with increased locomotion in CF).
DOI: https://doi.org/10.7554/eLife.32808.024

neurons (NPY, Hcrt, AgRP) and decreased numbers of anorexigenic neurons (POMCa/b, AVT) do not have a direct effect on food intake in Pachón CF larvae and juveniles. This is in contrast to adults and juveniles of other CF populations such as Tinaja or Molino, who carry a mutation in the coding sequence of the melanocortin four receptor (mc4r) (*Aspiras et al., 2015*), a constitutively-activated receptor on which AgRP acts as an antagonist, including in fish (*Cerdá-Reverter et al., 2011*). In the Pachón CF, who do not carry this mutation, compensatory mechanisms to the disequilibrium in orexigenic/anorexigenic neurons might exist. This may include modulations by other neurotransmitter systems which are also modified in CF, such as the serotonergic system (*Elipot et al., 2013*; *Elipot et al., 2014a*), well known as a negative appetite-regulating system through its action on the melanocortin system (*Lam et al., 2010*).

Hcrt in the hypothalamus has a conserved role in the regulation of locomotion and sleep (*Sakurai et al., 2010*). Here, we have developmentally manipulated the number of Hcrt neurons in CF through *Lhx9* MO knock-down. We found that 7dpf CF with reduced numbers of Hcrt neurons show decreased locomotion and increased hypoactivity state at larval stage, and are indistinguishable from the SF behavior. Therefore we propose that high locomotor activity in CF is due to developmental evolution of their Hcrt system (*Figure 7C*).

Mice with reduced numbers of orexin neurons are narcoleptic (*Hara et al., 2001*) and zebrafish overexpressing Hcrt are hyperactive and display an insomnia-like phenotype (*Prober et al., 2006*) (in fish, sleep-like states can be identified behaviorally as periods of inactivity associated with increased arousal thresholds). Moreover, in zebrafish, Hcrt seems to control the transitions from states of no or low activity into a high activity state (*Prober et al., 2006*). In *A. mexicanus*, high locomotor activity and reduction of sleep are described in 3 weeks old and adult CF (*Yoshizawa et al., 2015*; *Duboué et al., 2011*), and genetic or pharmacological inhibition of Hcrt signaling increases sleep duration in adult CF (*Jaggard et al., 2017a*). Although we have not directly measured sleep in the present study, it is tempting to speculate that developmental evolution of the Hcrt cluster in CF plays a role in sleep loss. Other neurotransmitters might play a role as well: noradrenalin, whose brain levels are higher in Pachón CF than in SF (*Elipot et al., 2014a*), also regulates sleep in CF (*Duboué et al., 2012*).

In conclusion, we provide a developmental origin to the evolution of behavior in cavefish. Juveniles and adults of Pachón CF show vibration-attraction behavior mediated by their more sensitive lateral line neuromasts (*Yoshizawa et al., 2010*), and ablation of the lateral line enhances sleep in the Pachón cavefish population (*Jaggard et al., 2017b*), suggesting that heightened sensory input participates to evolutionarily derived sleep loss in adults. QTL studies have recently demonstrated that increased locomotion/sleep loss and enhanced sensory responsiveness have a different genetic determinism (*Yoshizawa et al., 2015*). No loci linked to the CF locomotor activity phenotype have been identified so far, but they may well correspond to early embryonic signaling pathways which determine the numbers of neurons in the Hcrt hypothalamic cluster.

## Materials and methods

### Fish samples

Laboratory stocks of *A. mexicanus* surface fish and cavefish (Pachón population) were obtained in 2004 from the Jeffery laboratory at the University of Maryland, College Park, MD. Fish are maintained at 23–26°C on a 12:12 hr light:dark cycle. Embryos were collected after spawning and fixed at various stages in 4% paraformaldehyde (PFA), overnight at 4°C. Embryos younger than 24hpf that had not hatched were de-chorionated manually then fixed again for 12 hr at 4°C. After progressive dehydration in methanol, they were stored at −20°C. For morpholino injection experiments, eggs were obtained by in vitro fertilization. *A. mexicanus* development is highly similar to zebrafish in the first 20 hr post-fertilization (hpf) and, importantly, there is no difference in early developmental timing between the cave and surface forms (*Hinaux et al., 2011*). Animals were treated according to the French and European regulations for handling of animals in research. SR's authorization for use of animals in research including *Astyanax mexicanus* is 91–116 and Paris Centre-Sud Ethic Committee authorization numbers are 2012–0052, −0053, and −0054.

### Phylogenetic analyses

*Astyanax mexicanus* sequences similar to Pomcb, Pomca, AgRP, IT, CART3, NPY, AVT and Hcrt were obtained by TBlastN searches using *Danio rerio* sequences on the *A. mexicanus* EST assembly or developmental transcriptome contig sequences [available at http://genotoul-contigbrowser.toulouse.inra.fr:9099/index.html, (*Hinaux et al., 2013*)]. Alignments were constructed using available sequences retrieved by blast searches on public databases for a representative set of chordates. Sequences were aligned using Mafft (*Katoh and Standley, 2013*). Alignments were slightly corrected manually to eliminate major mistakes. Ambiguous regions were identified by visual inspection and removed manually. Maximum-likelihood (ML) analyses were performed using the PhyML program (*Guindon et al., 2003*), with the LG model of amino-acid substitution and a BioNJ tree as the input tree. A gamma distribution with four discrete categories was used in these ML analyses. The

gamma shape parameter and the proportion of invariant sites were optimized during the searches. The statistical significance of the nodes was assessed by bootstrapping (100 replicates). Resulting phylogenetic trees are not shown and are available upon request.

## cDNA cloning and data availability

Total RNA from cavefish embryos at multiple stages was reverse transcribed with random primers using AMV reverse transcriptase (Promega). Sequence for Hcrt transcript was obtained from our own next-generation sequencing data, and partial 750 bp sequence was amplified by PCR using specific primers. PCR products were subcloned in TOPO-PCR II vector (Invitrogen, Carlsbad, CA, USA) and sequenced. They corresponded to XM_007287820.3 (Hcrt precursor mRNA predicted from the genome). Pomcb, Pomca, AgRP, IT, AVT, CART3 and NPY cDNAs were obtained from our in house clonal library of ESTs (accession numbers FO375681, FO257910, FO289826, FO221370, FO234678, FO230154, FO263072). PCR products were sequenced by Sanger method before probe synthesis. Lhx9 (NM001291259.1) and Lhx7 cDNA were previously cloned by our group and by David Stock, respectively. These two LIM-homeodomain factors were chosen for analysis because unpublished work in the group had mapped their expression domains in the exact territories where Hcrt and NPY cells (modified in numbers in cavefish) are located.

## Whole-mount in situ hybridization

The expression patterns of nine neuropeptides (Agouti-related protein/AgRP, Arginine-Vasotocin/AVT, Cocaine and Amphetamine related transcript/CART3, Oxytocin-Isotocin/IT, Melanin-Concentrating Hormone/MCH, Neuropeptide Y/NPY, Hypocretin/Hcrt (also called orexin), and Pro-opiomelanocortin a and b/POMCa and POMCb) were systematically studied at three embryonic and larval stages: 24 hpf (hatching), 36 hpf (swimming larva), and 84 hpf (before first feeding) (*Hinaux et al., 2011*), which correspond to important steps in the morphogenesis and development of the secondary prosencephalon where most peptidergic neurons are located. These nine neuropeptides were chosen to cover the whole neuroendocrine territory in the basal forebrain (NPO, optic region, hypothalamus), and with respect to their known physiological roles in relation with known behavioral modifications in cavefish. The nine neuropeptides all show increasingly complex expression patterns as development proceeds. In some cases, earlier stages were also studied to determine the onset of neuropeptide expression.

cDNAs were amplified by PCR, and digoxygenin- or fluorescein-labelled riboprobes were synthesized from PCR templates. Embryos were rehydrated by graded series of EtOH/PBS, then for embryos older than 24 hpf, proteinase-K permeabilization at 37°C was performed (36hpf: 10 μg/ml, 15 min; 84 hpf: 40 μg/ml, 30 min; 7 dpf: 100 μg/ml, 45 min) followed by a post-fixation step. Riboprobes were hybridized for 16 hr at 65°C and embryos were incubated with anti-DIG-AP (Roche, dilution 1/4000) overnight at 4°C. Colorimetric detection with BCIP/NBT (Roche) was used. Mounted embryos were imaged on a Nikon Eclipse E800 microscope equipped with a Nikon DXM 1200 camera running under Nikon ACT-1 software. For fluorescent in situ hybridization, Cy3- and FITC-tyramides were prepared as described (*Zhou and Vize, 2004*). Embryos were incubated with anti-FITC-POD antibody (Roche, 1/400), washed in PBS/Tween 0.1% (PBST) and incubated for 20 min at room temperature with FITC-tyramide at 1/100. Tyramides were activated by $H_2O_2$ (Sigma, 0.003%) for 1 hr and washed again in PBST. The first peroxidase conjugate was inactivated by incubation in 3% $H_2O_2$ for 30 min at room temperature. Embryos were washed in PBST and incubated with the second antibody (anti-digoxygenin-POD, Roche, 1/200). The same protocol was applied for the Cy3-tyramide revelation. Then embryos were stained with Dapi at a final concentration of 1 μg/ml, overnight at 4°C, and washed in PBS before mounting. Fluorescent in situ hybridizations were imaged on a Leica-SP8 confocal microscope combined to Leica Application Suite software.

## Neuron quantification and statistical treatment

Neuron counting and delineation of expression domains were performed by eye on colorimetric in situ hybridizations, on dissected and mounted fish brains. Counts were performed under the microscope (not on pictures) while progressively changing the plane of focus and ticking each counted cell on the camera software. We found that this method was reliable and gave highly reproducible results (*Figure 1—figure supplement 3*). Investigators were not blinded to fish groups (CF vs SF or

control vs morphants) and animals were not randomly allocated to experimental groups because pigmentation renders SF and CF immediately distinguishable. In the absence of existing pilot studies, no effect size was pre-specified and therefore no sample size was predetermined. For pair-wise comparisons, and when the distribution of the data was not normal, we used Mann-Whitney non-parametric tests. For multiple comparisons, and when the distribution was normal, ANOVA analyses were performed, with Bonferroni's post-test. Statistical significance was set at $p < 0.05$ (*$p < 0.05$; **$p < 0.01$; ***$p < 0.001$). Values represent the results of at least two independent experimental replicates.

## Pharmacological treatments

Manually or chemically (Pronase 417 µg/mL; 3 min) de-chorionated CF embryos were incubated in 50 µM cyclopamine (C-8700, LC Laboratories) or 5 µM SU5402 (215543-92-3, Calbiochem) diluted in HBSS (55037C, Sigma) at developmental stages indicated in the main text. Controls were incubated in an equivalent concentration of ethanol or DMSO, respectively. They were washed in HBSS, raised in HBSS/methyl cellulose (M0387, Sigma)/Penicillin Streptomycin 1X (P4333, Sigma) and fixed at stages of interest. To define and ascertain the effect of cyclopamine and SU5402, we checked that hatched larvae have a typical 'comma shape' or tail bud defects, respectively.

## Morpholino and mRNA injections

FITC-labeled Morpholino oligos (MOs) were designed and produced by Gene Tools, complementary to the 5'UTR, AUG and exon-intron (e-i) junctions containing sequences in the *Lhx7* and *Lhx9* mRNA. The sequences were as follows:

*Lhx7 AUG MO*, 5'-GTTCATCTCTCCAGAACATGAGGGT-3';
*Lhx7 5'UTR MO*, 5'- ACTCAGGCTGAGCAACAGGAGAACC-3';
*Lhx7 e4-i4 splice blocking MO*, 5'-GTAAGTAATTCTGACCGTTCTTCCAT-3';
*Lhx9 AUG MO*, 5'-CCTTGCACCCCACCACTTCCATAC-3';
*Lhx9 5'UTR MO*, 5'-CTCCGCAGCCTCAGACCATCCGAAA-3';
*Lhx9 e1-i1 splice blocking MO*, 5'- GAAGTTAAAGATCTCACCGTCTCCC-3'.

MOs were re-suspended in distilled water to prepare stock solutions at a final concentration of 1 mM. Different dilutions of stock solutions were tested for each MO (working solutions, containing 0.05% Phenol red) to avoid abnormal phenotypes due to toxicity. Embryos were injected with 5–10 nL of working solutions using borosilicate glass pipettes (GC100F15, Harvard Apparatus LTD) pulled in a Narishige PN-30 Puller (Japan). For each experimental condition, the same concentration of a MO, with a similar sequence to the *Lhx9* 5'UTR MO, but containing five mismatches (5'-CgTTcCACCgCACCAgTTCgATAC-3', i.e. mm*Lhx9* MO), was injected as control. Injected embryos positive for FITC were screened under fluorescence.

Full length *Lhx7* and *Lhx9* cDNA were amplified (AccuPrime Pfx DNA Polymerase, ThermoFisher) and cloned into the pCS2 +vector using BamHI and Xbal restriction sites.

Primers used were as follows:

*Lhx7* forward, 5'-GGTGGTGGATCCACCATGTTCTGGAGAGATGAACAAACGG-3';
*Lhx7* reverse, 5'-ACCACCTCTAGACTAAAAATGCTCTGATAGGTTTGGGCT-3';
*Lhx9* forward, 5'-GGTGGTGGATCCACCATGGAAGTGGTGGGGGTGCAA-3';
*Lhx9* reverse, 5'-ACCACCTCTAGATTAGAAAAGATTTGTCAAGGTAGTTTGTGGG-3'; mRNAs were prepared in vitro with the mMessage mMachine kit (ThermoFisher) using the SP6 RNA polymerase. 5 µL of *Lhx7* or *Lhx9* mRNA were injected per embryo at a final concentration of 100–150 ng/µL and 150–200 ng/µL, respectively.

To test the efficiency of the splice blocking MOs, the E4E5 junction of *Lhx7* mRNA and E1E2 junction of *Lhx9* mRNA were amplified from cDNA samples and PCR products were visualized in an agarose gel (1%) to determine the sizes of the amplicons. As controls of size, genomic DNA was also included as template. The primers used were:

*Lhx7_* e4-e5 forward, 5'-AAGGAAACACCTACCACCTCG-3';
*Lhx7_* e4-e5 reverse, 5'-TGGTTCGTGCTCTTTTGGCT-3';
*Lhx9_*e1-e2 forward, 5'-CTGGCCAAAACGGTCCAGAT-3';
*Lhx9_* e1-e2 reverse, 5'-CGACTCCAAGGCCAGTTTACA-3'.

As control, levels of *β-actin* and *gapdh* mRNAs were also assessed through semi-quantitative RT-PCR. The primers used were: *gapdh* forward, 5'-GTTGGCATCAACGGATTTGG-3'; *gapdh* reverse, 5'-CCAGGTCAATGAAGGGGTCA-3';
  *β-actin* forward, 5'-GGATTCGCTGGAGATGATGCTC-3';
  *β-actin* reverse, 5'-CGTCACCTTCACCGTTCCAGT-3'.

## Behavioral analyses

For the feeding assays, 10 CF and 10 SF larvae were individually raised in 60 mm diameter Petri dishes containing 15 mL of blue water (*Elipot et al., 2014*) from 2.5 dpf onwards, in ambient light conditions. Twice a week, at noon, starting from 5.5dpf (when the mouth is opened) and until 1mpf, a precise excess amount of 2-day-old artemia nauplii was manually counted and added to each Petri dish (1st week: 100 nauplii, 2nd week: 120 nauplii, 3rd week: 200 nauplii and 4th week: 300 nauplii). Water volume was constant during 1 month including during the feeding test (water volume added with nauplii was removed immediately). Fish were allowed to feed for 5 hr. The amount of leftover nauplii was then removed and counted to deduce the number of nauplii eaten. On all other days, larvae were fed once a day with an excess of artemia nauplii and the medium was changed twice a day.

For the locomotion/activity assays, 6 dpf fish were individually placed in one well of a 24-well plate the day before starting the experiment, for habituation during 18–24 hr. For automatic tracking (ViewPoint imaging software), animals were illuminated from below with an infra-red light source and recorded with a Dragonfly2 camera (ViewPoint) during 26 hr. In order to simulate circadian conditions, lights were turned on from 8:30 (AM) to 22:30 (14:10 hr light:dark cycle). Average and minimum speed (mm/s) were calculated on 10 min bouts in CF to set a threshold speed (two times the minimum speed). Values under this threshold were considered as states of low-activity. After the recordings, animals were sacrificed and fixed for ISH processing and counting of Hcrt neurons.

## Acknowledgements

We thank Stéphane Père and Victor Simon for care of our *Astyanax* breeding colony, and Cynthia Froc of the Amatrace platform for help with behavior experiments. Work supported by an ANR (Agence Nationale pour la Recherche) grant [BLINDTEST] and an « Equipe FRM » grant [DEQ20150331745] from the Fondation pour la Recherche Médicale to SR.

## Additional information

### Funding

| Funder | Grant reference number | Author |
| --- | --- | --- |
| Agence Nationale de la Recherche | Blindtest | Sylvie Retaux |
| Fondation pour la Recherche Médicale | DEQ20150331745 RETAUX | Sylvie Retaux |

The funders had no role in study design, data collection and interpretation, or the decision to submit the work for publication.

### Author contributions

Alexandre Alié, Conceptualization, Investigation, Writing—original draft; Lucie Devos, Conceptualization, Investigation; Jorge Torres-Paz, Conceptualization, Investigation, Methodology, Writing—original draft; Lise Prunier, Fanny Boulet, Maryline Blin, Yannick Elipot, Investigation; Sylvie Retaux, Conceptualization, Supervision, Funding acquisition, Investigation, Writing—original draft, Project administration, Writing—review and editing

### Author ORCIDs

Sylvie Retaux (iD) http://orcid.org/0000-0003-0981-1478

## Ethics

Animal experimentation: Animals were treated according to the French and European regulations for handling of animals in research. SR's authorization for use of animals in research including Astyanax mexicanus is 91-116 and Paris Centre-Sud Ethic Committee authorization numbers are 2012-0052, -0053, and -0054.

## Decision letter and Author response

Decision letter https://doi.org/10.7554/eLife.32808.044
Author response https://doi.org/10.7554/eLife.32808.045

# Additional files

## Supplementary files

• Supplementary file 1. Cell counts. Table showing raw data for cell counts for the nine studied neuropeptides in CF and SF at three different developmental stages.
DOI: https://doi.org/10.7554/eLife.32808.025

• Transparent reporting form
DOI: https://doi.org/10.7554/eLife.32808.026

## Major datasets

The following previously published datasets were used:

| Author(s) | Year | Dataset title | Dataset URL | Database, license, and accessibility information |
|---|---|---|---|---|
| Hinaux H, Poulain J, Da Silva C, Noirot C, Jeffery W.R, Casane D, Retaux S | 2012 | FO375681 Astyanax mexicanus whole embryos and larvae neurula to swimming larvae Astyanax mexicanus cDNA clone ARA0AHA20YB20, mRNA sequence | https://www.ncbi.nlm.nih.gov/nucest/FO375681 | Publicly available at NCBI (accession no: FO375681). |
| Hinaux H, Poulain J, Da Silva C, Noirot C, Jeffery W.R, Casane D, Retaux S | 2012 | FO257910 Astyanax mexicanus whole embryos and larvae neurula to swimming larvae Astyanax mexicanus cDNA clone ARA0ABA19YH22, mRNA sequence | https://www.ncbi.nlm.nih.gov/nucest/FO257910 | Publicly available at NCBI (accession no: FO257910). |
| Hinaux H, Poulain J, Da Silva C, Noirot C, Jeffery W.R, Casane D, Retaux S | 2012 | FO289826 Astyanax mexicanus whole embryos and larvae neurula to swimming larvae Astyanax mexicanus cDNA clone ARA0ABA97YN12, mRNA sequence | https://www.ncbi.nlm.nih.gov/nucest/FO289826 | Publicly available at NCBI (accession no: FO289826). |
| Hinaux H, Poulain J, Da Silva C, Noirot C, Jeffery W.R, Casane D, Retaux S | 2012 | FO221370 Astyanax mexicanus whole embryos and larvae neurula to swimming larvae Astyanax mexicanus cDNA clone ARA0AAA24YM17, mRNA sequence | https://www.ncbi.nlm.nih.gov/nucest/FO221370 | Publicly available at NCBI (accession no: FO221370). |
| Hinaux H, Poulain J, Da Silva C, Noirot C, Jeffery W.R, Casane D, Retaux S | 2012 | FO234678 Astyanax mexicanus whole embryos and larvae neurula to swimming larvae Astyanax mexicanus cDNA clone ARA0AAA59YC09, mRNA sequence | https://www.ncbi.nlm.nih.gov/nucest/FO234678 | Publicly available at NCBI (accession no: FO234678). |
| Hinaux H, Poulain J, Da Silva C, Noirot C, Jeffery W.R, Casane D, Retaux S | 2012 | FO230154 Astyanax mexicanus whole embryos and larvae neurula to swimming larvae Astyanax mexicanus cDNA clone ARA0AAA6YH03, mRNA sequence | https://www.ncbi.nlm.nih.gov/nucest/FO230154 | Publicly available at NCBI (accession no: FO230154). |
| Hinaux H, Poulain J, Da Silva C, Noirot C, Jeffery W.R, Casane D, Retaux S | 2012 | FO263072 Astyanax mexicanus whole embryos and larvae neurula to swimming larvae Astyanax mexicanus cDNA clone ARA0ABA37YE05, mRNA sequence | https://www.ncbi.nlm.nih.gov/nucest/FO263072 | Publicly available at NCBI (accession no: FO263072). |

| Wes Warren, Su-zanne McCaugh | 2017 | PREDICTED: Astyanax mexicanus hypocretin neuropeptide precursor (hcrt), mRNA | https://www.ncbi.nlm.nih.gov/nucleotide/XM_007287820 | Publicly available at NCBI (accession no: XM_007287820.3). |

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
