## [Decision Letter]

[Editors’ note: a previous version of this study was rejected after peer review, but the authors submitted for reconsideration. The first decision letter after peer review is shown below.]

Thank you for submitting your work entitled "Developmental evolution of the forebrain in cavefish: from natural variations in neuropeptides to behavior" for consideration by *eLife*. Your article has been reviewed by three peer reviewers, and the evaluation has been overseen by a Senior/Reviewing Editor. The reviewers have opted to remain anonymous.

Our decision has been reached after consultation between the reviewers. Based on these discussions and the individual reviews below, we regret to inform you that your work will not be considered further for publication in *eLife*.

Specifically, the reviewers find the manuscript interesting and that it has potential. However, they also raised a number of concerns about essential missing controls and better characterization of the sleep phenotype. It is the policy of *eLife* to reject papers that need revisions that would require more than two months to complete. However, if you feel you can address the comments of the reviewers in the future, we would be happy to receive a new submission and every effort would be made to return the paper to the previous reviewers. Please see the full reviews below, which I hope you find useful, for details that would need to be addressed.

Reviewer #1:

The manuscript "developmental evolution of the forebrain in cavefish: from natural variations in neuropeptides to behavior" by Alié, Devos, Torres-Paz et al. studies whether differences in the neurodevelopment of the cave and surface fish populations of *Astyanax* underlie known behavioral adaptations of these fish. Using a tour de force whole mount in situ approach, the authors are able to show that certain neuropeptidergic cell groups (NPY, Hypocretin, AgRP neurons) are over-represented in cavefish larvae while others (POMC and AVT) were more abundant in surface fish. Importantly, the authors move beyond a pure descriptive work and start to dissect the regulatory network underlying the morphological changes in the brain. Using morpholino and chemical inhibitors the authors provide evidence that an expansion of SHH and Fgf8 in the forebrain of cavefish leads to increased lhx7/9 expression which in turn leads to higher numbers of NPY and Hcrt neurons. Most importantly and excitingly the authors are able to link these neuronal changes to the sleep behavior changes in cavefish which makes the study stand out and I recommend publication in *eLife* if the major comments listed below are addressed.

1) While I compliment the authors on the use of functional studies in an emerging model organism and the use of morpholino is acceptable in the absence of fully established gene editing methods, the morpholino experiments lack appropriate and necessary controls. First, it is unclear to me why the authors do not use a splice morpholino (do they assume a maternal contribution to be important in the phenotypes?). A splice morpholino has the big advantage of the simple ability to show that the morpholino is reducing the available amount of functional protein via PCR. If a splice morpholino cannot be used then a dose response curve to the morpholino on the protein level is necessary (which arguably is more complicated but needed). Furthermore, it is common practice to show specificity of the morpholino by rescuing the phenotype with RNA-co-injection.

2) The result that morpholino treated cavefish sleep more and this phenotype correlates with the number of Hcrt neurons is extremely exciting. I am just a little bit concerned about the methods of sleep recording. In previous studies that have recorded sleep in *Astyanax* larvae the methods were as follows: "Following an 18 hr acclimation period, fry were recorded for a period of 24 or 48 hr" (copied from Duboue, Keene and Borowsky, 2011). In this study the authors used only 1 hour acclimation and 1 hour recording which compared to the original study seems a bit short. In addition, Duboue, Keene and Borowsky, 2011 uses: "fry […] between 21 and 24 days old", while the authors used 7dpf old fish. A comparison between the two different ages (in surface and cavefish) is needed to show that such sleep behavior experiments can also be performed at these earlier time points as larval behavior is very different between 7dpf and 21 dpf.

3) Statistics for the neuron quantification. The authors depict the data as "the mean +/- SEM of at least two independent replicates". I have two general problems with this: First, two replicates are not enough to account for potential variation in whole mount in situ experiments (which can be very variable). Second the use of mean +/- SEM is the weakest way of displaying data and generally should be avoided.

*Reviewer #2:*

In "Developmental evolution of the forebrain in cavefish: from natural variations in neuropeptides to behavior" the authors seek to quantify cell number differences in different neuropeptidergic cell types between cave and surface fish. They then attempt to link early signaling to these differences in cell number to changes in behavior that have evolved in cavefish. Establishing these links would be an important step in understanding the evolution of behavior in this species, and more generally, as little is known about the genetic, developmental and neural bases of behavioral evolution in vertebrate species. However, there are a number of substantive concerns about this work that need to be addressed prior to coming to conclusions about these links.

First, the paper relies heavily on quantifying neuron populations using colormetric in situ hybridization. I find this potentially problematic. How do the authors know that they are comparing the same plane in cavefish and surface fish for accurate quantification? As most of the conclusions of the paper rest on there being differences in numbers between certain cell types, it is critical that these cell numbers are quantified accurately. Quantifications would be more convincing if they were performed on three dimensional images that did not rely on identifying precisely the same field of view and depth in each embryo. This could be done by confocal imaging of fluorescent in situs.

Second, linking gene expression during development to different numbers of cells of a certain type and to behavior relies on morpholino experiments. However, aside from a morpholino with mismatches relative to a single other morpholino they inject, they have not provided controls that convincingly demonstrate the specificity of their knock downs. At a minimum, rescue experiments by expressing an mRNA not targeted by the morpholino should be performed. Additional controls that would make these experiments more convincing would include morpholinos that are splice blocking, and assessing activity by RT-PCR and/or designing morpholinos that would target these genes in *Astyanax* but not in zebrafish, and demonstrating that they do not have the same effects when injected into zebrafish, indicating specificity of the morpholino. The ideal experiment would be to compare the effects of these morpholinos to a fish mutant for these genes. This is now possible in *Astyanax*, as genome editing technologies have been used in *Astyanax*, but is likely beyond the scope of this publication.

Third, the sleep analysis included in the paper needs to be greatly expanded to be convincing. Sleep assays were performed for one hour, and it was not specified what time of day or night, or if they were all performed at the same time of day, all of which is critical to assessing sleep. Furthermore, it is not standard, to my knowledge, to assess sleep for only an hour, and assessing sleep over at least a 24 hour period including one light and one dark cycle would be more in line with previous publications, and more convincing. Additionally, sleep assays were performed on 7 dpf embryos, presumably because the effects of morpholinos are transient. However, the authors use periods of over 60s to define sleep. This time period was defined as sleep in previous studies. However, these studies used fish that were 21-24 dpf. Is there evidence that length of sleep bouts are the same at these different stages of development? If not, determining what period of inactivity counts as sleep at 7dpf, using methods as in Duboue et al. 2011, is critical prior to assessing sleep at this stage.

Furthermore, the authors find an increase in sleep in morphant fish compared to controls. However, as they have not measured the overall activity of the fish, it is impossible to know whether their morphants have a change in sleep, or a change in overall activity, which is affecting the amount of time they are being still in the hour assessed.

Finally, they have not demonstrated that sleep is in fact different between cave and surface fish at this stage. Including surface fish in this analysis is key. As identifying changes in behavior is part of the novelty of this work, robustly assessing sleep behavior is critical to this manuscript.

Reviewer #3:

Alie et al. reports an interesting difference in the number of several neuropeptidergic cells in the hypothalamus between surface-dwelling (SF) vs. cave-dwelling form (CF)of *Astyanax mexicanus*. More specifically cave-dwelling fish possess more NPY, Hcrt, AgRP neurons and fewer POMCa, POMCb and VP neurons. To determine the developmental pathway responsible for the difference, the authors next examined the effect of manipulating Lhx7 or Lhx9 on the development of NPY or Hcrt neurons. Since all peptidergic cell types that show differences are located in the basal plate of the hypothalamus, the authors also examined the effect of Shh or Fgf8 pathway manipulation on the development of NPY or Hcrt neurons. Lastly, they examined feeding and sleeping behavior of the Lhx9 morphants.

*Astyanax mexicanus* is a powerful model organism to examine mechanisms underlying adaptation to extreme environment. I like this study for its developmental evolutionary approach and the authors attempt to place molecular evolution in a larger adaptive behavioral context. However while seeing the merit of the interesting observation they have made in the difference in the number of neuropeptidergic cells between the two forms, I fail to see how this work contributes significantly beyond this observation to provide advances in understanding either the mechanistic basis for this difference or behavioral significance of this difference.

1) The behavioral differences between SF and CF forms are striking and it would have been very interesting to learn about what mediates these behavioral differences. Locomotion, feeding and sleep are behaviors that are controlled and regulated by a large network of neurons that express different neuropeptides and neurotransmitters. I do not understand the author's logic of choosing a particular set of 9 neuropeptides, which have pleiotropic effects on many physiological process and diverse behavior. They also do not explain in the manuscript justification of their choice. To make any conclusion about "evolution of a neuronal system and the adaptive behavior," their strategy of choosing a few neurons with pleiotropic effects without considering a large network involved in a particular behavior is inadequate. The author's claim concerning behavior is solely based on the phenotype of Lhx9 mutant, which shows alteration in Hcrt neuron number as well as "sleep" phenotype. While involvement of Hcrt neuron in sleep control is well established, is the difference in Hcrt neuron number really the cause of the sleep pattern difference in these two forms when sleep can be also affected by NPY neurons, by change in circadian rhythm, other modulatory systems, hormones that affect arousal and stress, etc.?

2) I am not convinced that the study provides significant novel insight about the mechanism behind the development of these neurons. The role of Lhx9 on the Hcrt neurons is well known and observed in both mouse and zebrafish. It is not surprising that Lhx9 plays a role in Hcrt neuron development in cave fish. The fact that Shh and Fgf8 signaling variation early in development affects neuropeptidergic patterning in the cavefish brain adds to the large amount of existing data on the role of these two molecules on the development of hypothalamic cell types, yet do not represent a major advance in our understanding of the development of hypothalamic neuropeptidergic neurons.

3) The fact that Lhx9 morphants show sleep phenotype is consistent with the literature and is completely expected.

[Editors’ note: what now follows is the decision letter after the authors submitted for further consideration.]

Thank you for resubmitting your work entitled "Developmental evolution of the forebrain in cavefish, from natural variations in neuropeptides to behavior" for further consideration at *eLife*. Your revised article has been favorably evaluated by Marianne Bronner as the Senior and Reviewing Editor, and two reviewers.

The manuscript has been much improved but there are some remaining issues that need to be addressed before acceptance, as outlined below:

1) The authors still haven't addressed a main original concern – quantifying colorimetric in situ. They need to show that their method is accurate using at least a few controls in situ that are stained with both methods to show that the data is comparable and valid.

2) The authors should explain why the control morpholino (Lhx9 mm MO) fish have a higher activity level compared to control uninjected cavefish? No statistics are shown between uninjected cavefish and Lhx9 mm MO injected cavefish. Why do none of the fish have reduced activity in the dark?

3) Is the hyperactivity measure a standard measure? If so, citations should be included. What is the relationship to this number and amount of time spent sleeping and why isn't sleeping time quantified?

---

## [Author Response]

[Editors’ note: the author responses to the first round of peer review follow.]

Reviewer #1: […] 1) While I compliment the authors on the use of functional studies in an emerging model organism and the use of morpholino is acceptable in the absence of fully established gene editing methods, the morpholino experiments lack appropriate and necessary controls. First, it is unclear to me why the authors do not use a splice morpholino (do they assume a maternal contribution to be important in the phenotypes?). A splice morpholino has the big advantage of the simple ability to show that the morpholino is reducing the available amount of functional protein via PCR. If a splice morpholino cannot be used then a dose response curve to the morpholino on the protein level is necessary (which arguably is more complicated but needed). Furthermore, it is common practice to show specificity of the morpholino by rescuing the phenotype with RNA-co-injection.

Done. We have now performed the requested splice morpholino experiments, for both Lhx7 and Lhx9 knock-downs. In both cases, the PCR controls were conclusive. And for both genes, effects on numbers of neuropeptidergic neurons were found, confirming the effects previously found with ATG and 5’UTR morpholinos, and reinforcing the demonstration of the function of Lhx7 and Lhx9 in NPY and Hcrt neuron specification, respectively.

Moreover and conversely, novel Lhx7 and Lhx9 overexpression experiments by mRNA injection produces the opposite effects, i.e., an increase in the number of NPY and Hcrt cells, respectively. Finally, mRNA co-injection rescued the morpholino phenotypes, confirming the specificity of observed effects.

2) The result that morpholino treated cavefish sleep more and this phenotype correlates with the number of Hcrt neurons is extremely exciting. I am just a little bit concerned about the methods of sleep recording. In previous studies that have recorded sleep in Astyanax larvae the methods were as follows: "Following an 18 hr acclimation period, fry were recorded for a period of 24 or 48 hr" (copied from Duboue, Keene and Borowsky, 2011). In this study the authors used only 1 hour acclimation and 1 hour recording which compared to the original study seems a bit short. In addition, Duboue, Keene and Borowsky, 2011 uses: "fry […] between 21 and 24 days old", while the authors used 7dpf old fish. A comparison between the two different ages (in surface and cavefish) is needed to show that such sleep behavior experiments can also be performed at these earlier time points as larval behavior is very different between 7dpf and 21 dpf.

Done. We have performed completely new series of experiments. We now provide locomotion assays and analyses on 24 hours cycles. In addition, we now provide comparisons including surface fish, Lhx9 morphant cavefish, and control cavefish. The results show that the difference between surface fish and cavefish in very similar in 7dpf larvae and in 21dpf larvae, allowing us to use the same criteria as in previously published papers. Please see revised Figure 6 and corresponding text.

3) Statistics for the neuron quantification. The authors depict the data as "the mean +/- SEM of at least two independent replicates". I have two general problems with this: First, two replicates are not enough to account for potential variation in whole mount in situ experiments (which can be very variable). Second the use of mean +/- SEM is the weakest way of displaying data and generally should be avoided.

About replicates for HIS experiments: we agree that the HIS technique can be variable in staining intensity or in signal/background ratio from one experiment to another. However, in the present case we did not measure staining intensities; we counted positively-labelled cells, which do not depend on the intensity or dynamics of HIS revelation. Therefore we believe that two replicates of the experiments are sufficient. This notion is also supported by the small inter-individual variations observed in the results of cell counts.

About the use of mean +/- SEM: we agree. We have modified the presentation of the results and now provide box plots throughout the manuscript. This novel presentation confirms reproducible results from one experiment to another (see previous point). Moreover, it also shows, as known in the *Astyanax* community, that cavefish show more inter-individual variability than surface fish. Despite this, clear differences between the two morphs are observable and are statistically significant (Mann-Whitney non-parametric tests).

Reviewer #2:[…] However, there are a number of substantive concerns about this work that need to be addressed prior to coming to conclusions about these links.First, the paper relies heavily on quantifying neuron populations using colormetric in situ hybridization. I find this potentially problematic. How do the authors know that they are comparing the same plane in cavefish and surface fish for accurate quantification? As most of the conclusions of the paper rest on there being differences in numbers between certain cell types, it is critical that these cell numbers are quantified accurately.

Most of the neuronal clusters we have quantified are not in a single “plane”, but rather represent 3D shapes along the 3 axes of the embryonic brain. This is the reason why neurons counts have been performed under the microscope (not on still pictures) while progressively changing the plane of focus and ticking each counted cell on the camera software. Moreover, neurons counts have been performed after exhaustive anatomical analyses to characterize, at each studied stage, the exact localization of each neuronal cluster, by comparing lateral and ventral views, and by localizing each cluster with respect to anatomical landmarks such as ventricles or commissures or neuroepithelial shapes. Finally it should be noted that in several simple cases where only one cluster was present, there was absolutely no ambiguity on the counts. This way, we are certain that neuronal clusters have been accurately quantified.

Quantifications would be more convincing if they were performed on three dimensional images that did not rely on identifying precisely the same field of view and depth in each embryo. This could be done by confocal imaging of fluorescent in situs.

Another reviewer acknowledged we have used a “tour de force whole mount in situ approach”. The amount of work and time analysis using fluorescent in situs would have been even greater, therefore we choose colorimetric in situs. However, as now shown in Supplemental material in one specific example for Hcrt cells at 24hpf, the cell counts using the two types of revelation are exactly the same, and the variability is even lower with colorimetric in situ. Moreover, it should be noted that we did use fluorescent in situ in other parts of the manuscript, when needed and when appropriate, for example for interpreting the respective positions of different peptidergic clusters after double fluorescent staining.

Second, linking gene expression during development to different numbers of cells of a certain type and to behavior relies on morpholino experiments. However, aside from a morpholino with mismatches relative to a single other morpholino they inject, they have not provided controls that convincingly demonstrate the specificity of their knock downs. At a minimum, rescue experiments by expressing an mRNA not targeted by the morpholino should be performed. Additional controls that would make these experiments more convincing would include morpholinos that are splice blocking, and assessing activity by RT-PCR and/or designing morpholinos that would target these genes in Astyanax but not in zebrafish, and demonstrating that they do not have the same effects when injected into zebrafish, indicating specificity of the morpholino. The ideal experiment would be to compare the effects of these morpholinos to a fish mutant for these genes. This is now possible in Astyanax, as genome editing technologies have been used in Astyanax, but is likely beyond the scope of this publication.

Done. We have now performed the additional requested splice morpholino experiments, for both Lhx7 and Lhx9 knock-downs. In both cases, the PCR controls were conclusive. And for both genes, effects on numbers of neuropeptidergic neurons were found, confirming the effects previously found with ATG and 5’UTR morpholinos, and reinforcing the demonstration of the function of Lhx7 and Lhx9 in NPY and Hcrt neuron specification, respectively.

Moreover and conversely, novel Lhx7 and Lhx9 overexpression experiments by mRNA injection produce the opposite effects, i.e., an increase in the number of NPY and Hcrt cells, respectively. Finally, mRNA co-injection rescued the morpholino phenotypes, confirming the specificity of observed effects.

Third, the sleep analysis included in the paper needs to be greatly expanded to be convincing. Sleep assays were performed for one hour, and it was not specified what time of day or night, or if they were all performed at the same time of day, all of which is critical to assessing sleep. Furthermore, it is not standard, to my knowledge, to assess sleep for only an hour, and assessing sleep over at least a 24 hour period including one light and one dark cycle would be more in line with previous publications, and more convincing.

Done, we have now performed assays on 24 hour cycles.

Additionally, sleep assays were performed on 7 dpf embryos, presumably because the effects of morpholinos are transient. However, the authors use periods of over 60s to define sleep. This time period was defined as sleep in previous studies. However, these studies used fish that were 21-24 dpf. Is there evidence that length of sleep bouts are the same at these different stages of development? If not, determining what period of inactivity counts as sleep at 7dpf, using methods as in Duboue et al. 2011, is critical prior to assessing sleep at this stage. Furthermore, the authors find an increase in sleep in morphant fish compared to controls. However, as they have not measured the overall activity of the fish, it is impossible to know whether their morphants have a change in sleep, or a change in overall activity, which is affecting the amount of time they are being still in the hour assessed.

Done. We have now performed completely new series of experiments. We now provide locomotion/activity assays and analyses on 24 hours cycles. In addition, we now provide comparisons including surface fish, cavefish, Lhx9 morphant cavefish, and control cavefish. The results show that the difference between surface fish and cavefish in very similar in 7dpf larvae and in 21dpf larvae, allowing us to use the same criteria as in previously published papers. Please see revised Figure 6 and corresponding text.

Finally, they have not demonstrated that sleep is in fact different between cave and surface fish at this stage. Including surface fish in this analysis is key. As identifying changes in behavior is part of the novelty of this work, robustly assessing sleep behavior is critical to this manuscript.

Done, we have now included surface fish in the additional experiments, which are compared to Lhx9 morphant cavefish, control mismatch MO cavefish, and un-injected cavefish.

Reviewer #3:[…] However while seeing the merit of the interesting observation they have made in the difference in the number of neuropeptidergic cells between the two forms, I fail to see how this work contributes significantly beyond this observation to provide advances in understanding either the mechanistic basis for this difference or behavioral significance of this difference.1) The behavioral differences between SF and CF forms are striking and it would have been very interesting to learn about what mediates these behavioral differences. Locomotion, feeding and sleep are behaviors that are controlled and regulated by a large network of neurons that express different neuropeptides and neurotransmitters. I do not understand the author's logic of choosing a particular set of 9 neuropeptides, which have pleiotropic effects on many physiological process and diverse behavior. They also do not explain in the manuscript justification of their choice. To make any conclusion about "evolution of a neuronal system and the adaptive behavior," their strategy of choosing a few neurons with pleiotropic effects without considering a large network involved in a particular behavior is inadequate. The author's claim concerning behavior is solely based on the phenotype of Lhx9 mutant, which shows alteration in Hcrt neuron number as well as "sleep" phenotype. While involvement of Hcrt neuron in sleep control is well established, is the difference in Hcrt neuron number really the cause of the sleep pattern difference in these two forms when sleep can be also affected by NPY neurons, by change in circadian rhythm, other modulatory systems, hormones that affect arousal and stress, etc.?

I am sorry but I don’t really follow the reviewer’s reasoning or what he/she would prefer as an approach for our study. On one hand he/she says that feeding or sleep behaviors are regulated by large networks of neurons expressing multiple neurotransmitters; and on the other hand, he/she says that involvement of Hcrt neurons (i.e. on cell group, one neurotransmitter) in sleep control is well established.

Our approach was through the window of the developmental evolution of the basal forebrain and its fine neuronal anatomy. We have first studied a large set of 9 neuropeptidergic cell types, and then, according to the results of the comparative developmental neuroanatomy, we have focused on 2 specific neurotransmission systems that we thought were relevant to the known behavioral phenotypes of cavefish. And indeed, through manipulation of the development of this neuronal cluster, we also affect the “candidate” sleep behavior. The originality of this work is that it is performed in an evolutionary context. Of course we do not claim that we have discovered a role for Hcrt in sleep regulation. We propose a developmental mechanism at work in a “natural mutant”, to explain the evolution of its anatomy and related behavior. Our results on the role of Lhx9/Hcrt hold true without having to rule out an influence of other factors.

2) I am not convinced that the study provides significant novel insight about the mechanism behind the development of these neurons. The role of Lhx9 on the Hcrt neurons is well known and observed in both mouse and zebrafish. It is not surprising that Lhx9 plays a role in Hcrt neuron development in cave fish. The fact that Shh and Fgf8 signaling variation early in development affects neuropeptidergic patterning in the cavefish brain adds to the large amount of existing data on the role of these two molecules on the development of hypothalamic cell types, yet do not represent a major advance in our understanding of the development of hypothalamic neuropeptidergic neurons.

Although we agree that Lhx9/Hcrt is not novel, we want to emphasize that Lhx7/NPY link is completely novel and an original finding of our study. In the cover letter, we wrote: We are aware that our study does not cover the exact detailed mechanisms by which developmental variations in hypothalamic networks are achieved (receptors involved, signaling pathways, transcriptional mechanisms…), but we believe that these issues should rather be studied in model organisms like zebrafish for which powerful genetic tools are available.

Finally and again, the novelty of our findings comes from the implication of the studied developmental pathways and genes in the *evolution* of the brain, not from their functional role in the development of this or that neuronal type.

3) The fact that Lhx9 morphants show sleep phenotype is consistent with the literature and is completely expected.

Cavefish morphants for Lhx9 do not exactly show a “sleep phenotype”, they rather have a rescued sleep phenotype: they sleep more than normal cavefish, in a comparable manner to what surface fish do.

Yes, knocking-down Lhx9 was expected from the literature to modify sleep. The originality of the work is that by manipulating this developmental pathway, we rescue an evolutionary phenotype, which is the best we can achieve from the evo-devo perspective. Our study is, to our knowledge, the first or one of the first to accomplish this goal and to fill the gap in a functional evo-devo perspective.

[Editors' note: the author responses to the re-review follow.]

The manuscript has been much improved but there are some remaining issues that need to be addressed before acceptance, as outlined below:1) The authors still haven't addressed a main original concern – quantifying colorimetric in situ. They need to show that their method is accurate using at least a few controls in situ that are stained with both methods to show that the data is comparable and valid.

We are sorry but we believe we have provided such controls, which are indeed important, in Figure 1—figure supplement 3. This figure shows comparison of colorimetric versus fluorescent in situ labeling for Hcrt at 24hpf, in SF and CF (chosen because Hcrt/24hpf is probably one of the most important neuropeptide and stage in our story). It clearly demonstrates that cell counts are identical for a given morph with the two methods, and that the difference between the two morphs is also detectable and identical with the two methods. To further convince the reviewer on the quantification method, we now provide additional “videos” to travel through stacks of photographs that were used to count the cells in the depth of the brain. This shows that the quantification was accurate, as the reproducibility of the results from one experiment to another also attests.

If the reviewer is asking for double labeling (colorimetric + fluorescent) on the same sample, then it is simply not possible to provide this: the purple precipitate of the colorimetric in situ would mask the fluorescent signal of the fluorescent in situ, and thus prevent any correct counting of the cells.

2) The authors should explain why the control morpholino (Lhx9 mm MO) fish have a higher activity level compared to control uninjected cavefish? No statistics are shown between uninjected cavefish and Lhx9 mm MO injected cavefish.

It is true that the Lhx9 mm MO curves and values are slightly higher than the control un-injected CF, but the differences are statistically non-significant (ANOVA and Mann-Whitney non-parametric tests, performed exactly in the same way as for comparisons between other conditions). That is the reason why “no statistics” were shown. All tests have been redone to write this response, and confirmed the absence of significance between the two conditions. To clarify this on the graphs, we have now added “ns” in Figure 6 and Figure 6—figure supplement 1.

Why do none of the fish have reduced activity in the dark?

This is indeed an intriguing question, but we have no answer to this. However, it should be noted that the activity curves provided in our manuscript are representative and consistent with previous publications on locomotor activity in *A. mexicanus*. For example, please check Figure 1AB of Yoshizawa et al. BMC Biology (2015), showing locomotor activity in SF and CF adults. The 24h curves are “identical” to our curves on larvae. The curves look so flat that the authors did not even bother to quantify locomotion during day versus night, and they pooled activity over 24 hours periods.

Moreover, we have unpublished data in the lab, on other projects, confirming the representativeness of the curves provided in the present manuscript at larval and juvenile stages.

Because this is an intriguing aspect, we have now added a sentence: “Of note and as reported also for adults [Yoshizawa et al., 2015], the locomotion did not vary significantly according to the day /night periods”.

3) Is the hyperactivity measure a standard measure? If so, citations should be included. What is the relationship to this number and amount of time spent sleeping and why isn't sleeping time quantified?

(We assume there is a typo in the question and the reviewer wanted to write “hypoactivity” instead of hyperactivity).

Sleeping time was not quantified because this would have implied a real study on sleep in *Astyanax* 7dpf larvae, which we believe is out of the scope of this paper, and for which we do not have the equipment and facility to perform the necessary experiments. This is the reason why we have focused exclusively on locomotor activity in the Results section. In the Discussion as well, we have now re-written the paragraph discussing this aspect, to avoid any over interpretation of the data:

“Hcrt in the hypothalamus has a conserved role in the regulation of locomotion and sleep [Elipot et al., 2014]. […] Other neurotransmitters might play a role as well: noradrenalin, whose brain levels are higher in Pachón CF than in SF [66], also regulates sleep in CF [Prober et al., 2006].”

Concerning “hypoactivity”, as explained in Materials and methods, low activity was defined as follows: “Average and minimum speed (mm/sec) were calculated on 10 minutes bouts in CF to set a threshold speed (2 times the minimum speed). Values under this threshold were considered as states of low-activity.” Hypoactivity is defined in the zebrafish catalog of behaviors (Towards a Comprehensive Catalog of Zebrafish Behavior 1.0 and Beyond; Kalueff et al. 2013, Zebrafish). As an example, Mathuru et al., studying behavioral responses to odorant molecules (Current Biology, 2012) used this parameter, defined and calculated similarly to the present paper (“…episodes were defined as episodes greater than one second in duration during which the swim speed never exceeded half the mean baseline speed”). Citations are now included.

Because we did not analyze sleep per se, we had used this parameter as a proxy of the fish “resting time”. It appears to be inversely correlated to locomotion in the different experimental conditions (SF, CF, Lhx9 MO and mmMO), and the important point is that for this parameter also, the Lhx9 CF morphants behave the same way as the SF, implying a role for the Hcrt neurons.

We hope the reviewer will be satisfied by these changes.